# Scaling Knowledge Editing in LLMs to 100,000 Facts with Neural KV Database

**Weizhi Fei**[1]   **Hao Shi**[1]   **Jing Xu**[2]   **Jingchen Peng**[1]   **Jiazheng Li**[3] **Jingzhao Zhang**[2,*]

**Bo Bai**[4]   **Wei Han**[4]   **Zhenyuan Chen**[4]   **Xueyan Niu**[4,*]

[1]Department of Mathematical Sciences, Tsinghua University
[2]IIIS, Tsinghua University
[3]College of Artificial Intelligence, Tsinghua University
[4]Huawei Technologies Co., Ltd.
[*]Corresponding author

```
{fwz22, shih22, pjc22}@mails.tsinghua.edu.cn
{xujing21, jingzhaoz}@mails.tsinghua.edu.cn
foreverlasting1202@outlook.com
{baibo8, harvey.hanwei, chenzhenyuan, niuxueyan3}@huawei.com
```

## Abstract

Efficiently editing knowledge stored in Large Language Models (LLMs) enables model updates without large-scale training. One promising solution is Locate-and-Edit (L&E), allowing simultaneous modifications of a massive number of factual knowledge. However, such editing may compromise the general abilities of LLMs and even result in forgetting edited facts when scaling up to thousands of edits. In this paper, we model existing linear L&E methods as querying a Key-Value (KV) database. From this perspective, we then propose NeuralDB, an editing framework that explicitly represents the edited facts as a neural KV database equipped with a non-linear gated retrieval module. With simple modification over L&E methods, our framework not only significantly extends the capacity of knowledge editing but also eliminates the associated side effects. Comprehensive experiments involving the editing of 10,000 facts were conducted on the ZsRE and Counter-Fact datasets, including GPT2-XL, GPT-J (6B) and Llama-3 (8B). The results demonstrate that NeuralDB excels in all metrics of editing success while maintaining original performance evaluated by six representative text understanding and generation tasks. Further experiments indicate that NeuralDB maintains its effectiveness even when scaled to 100,000 facts (**50×** more than in prior work).

## 1 Introduction

Updating the knowledge stored in the parameters of Large Language Models (LLMs) is crucial to refreshing outdated information (Zhu et al., 2024) and integrating domain-specific knowledge to facilitate customization (Ge et al., 2023). However, retraining LLMs from scratch is often impractical due to the substantial computational resources and time required. Fine-tuning, while a more feasible approach, can lead to catastrophic forgetting (Luo et al., 2023; Gekhman et al., 2024). To address these challenges, *knowledge editing* (KE) methods (Wang et al., 2024a; Mitchell et al., 2022a; Zheng et al., 2023a) have emerged as promising solutions that enable precise and cost-effective modifications of specific factual associations within LLMs.

Editing massive knowledge is an important but challenging task in KE. *Locate-and-Edit* (L&E) methods (Meng et al., 2022; Li et al., 2024b) are the main solutions to achieve this goal. The L&E paradigm trains the specific activation residual for new facts and incorporates the learned activations into the target parameter $\mathbf{W}$ by introducing a linear perturbation $\Delta$. Additionally, to ensure that the activation of "general knowledge" remains unmodified, these methods sample extensive data from Wikipedia representing the general knowledge. The modified parameters $\Delta$ are then determined by solving least squares problems to ensure that the residuals required are generated for

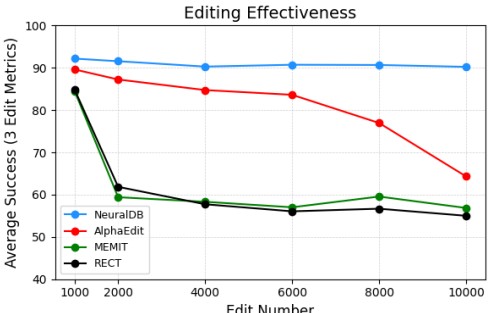 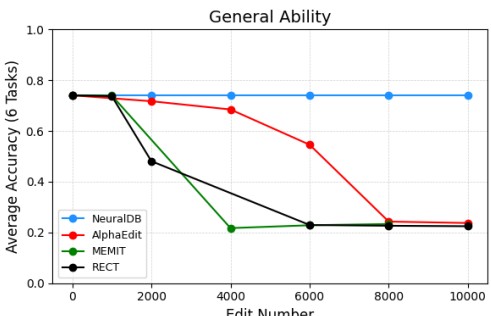

Figure 1: **The proposed NeuralDB scales edited facts up to** $10,000$ **without losing general abilities.** *Left:* Average of efficacy, generalization, and specificity. *Right:* Average performance on tasks (MMLU, SciQ, Commonsense QA, ARC Challenge, Lambada, WSC273).

the new facts without affecting the activation of sampled knowledge (Meng et al., 2022). Notably, AlphaEdit (Fang et al., 2025) introduces null space projection to enhance the preservation of the sampled knowledge, effectively maintaining the general capabilities of LLMs.

We refer to the *capacity* of KE methods as the maximum number of edited facts that can be handled without compromising general capabilities. Despite great progress, the capacity of existing methods is limited to hundreds of facts. When editing thousands of facts, the post-edited models often experience a decline in the valuable general abilities developed through extensive training (see Fig. 1). This decline can be attributed to the inadequacy of the sampled subset from Wikipedia in representing general capabilities. Additionally, previously updated information tends to fade as more facts are edited, due to the suboptimal capacity of the linear systems employed by these editing methods.

In this paper, we find that most L&E methods, including MEMIT (Meng et al., 2023), D4S (Huang et al., 2024), and AlphaEdit (Fang et al., 2025), can be understood from the perspective of the Key-Value (KV) database. We conceptualize these methods as querying a KV database, wherein a certain hidden state serves as the query to retrieve the corresponding learned residual. Formally, the updated parameters of these methods can be interpreted as a weighted average of the residual matrix associated with the edited facts. We empirically investigate these weights across multiple post-edited models and show that they exhibit an extremely sparse form during inference. Specifically, when inferring on the edited fact, the weights closely resemble a one-hot vector, with only the weight corresponding to the edited fact being non-zero, thereby returning the associated residual. Conversely, when inference is made on unrelated content, the weights are in the form of a zero vector, thereby preventing interference.

In light of this novel perspective, we propose a Neural KV Database (NeuralDB) editing framework, which integrates the target FFN layer with a gated non-linear retrieval module that replaces the original linear perturbation $\Delta$. This non-linear retrieval module overcomes the limitations of linearity in the L&E methods and enjoys greater capacity. With the gated mechanism, our method can both protect the general ability and reduce the computation costs from sampling Wikipedia. Additionally, NeuralDB is easy to manage for supporting operations such as appending, modifying, and deleting. We provide the quantitative and qualitative evidence of supporting delete operation in Appendix J.5.

To validate the capacity of NeuralDB, we conducted comprehensive experiments on two KE benchmarks across three models: GPT-2 XL (Radford et al., 2019), GPT-J (6B)(Wang, 2021), and Llama-3-Instruct (8B)(Grattafiori et al., 2024). Our results demonstrate several advantages of NeuralDB:

(i) NeuralDB achieves significantly improved performance in editing success across various metrics, while preserving fluency and consistency in post-edited models, particularly for 10,000 edited facts.

(ii) After editing 10,000 facts, NeuralDB maintains the quality of the generated text in Llama-3-8B across six widely adopted text understanding and generation tasks.

(iii) Extensive scaling experiments with 100,000 edited facts ($50\times$ more than AlphaEdit (Fang et al., 2025)) further demonstrate the high capacity of NeuralDB.

This combination of large capacity without a loss in generation quality underscores the potential of NeuralDB for trustworthy and customizable deployment of language models.

## 2 BACKGROUND

Current KE methods typically focus on updating transformer-based LLMs with factual knowledge that can be represented as triple $(s, r, o)$, where $s$ denotes the subject, $r$ represents the relational predicate, and $o$ is the object. For instance, the fact "The latest World Cup was held in Qatar." can be represented as ("The latest World Cup", "was located in", "Qatar"). Conversely, triples can be transformed into natural language sentences, and we treat these two representations as interchangeable in the sequel. We denote the edited facts as a set of revised tuples $\mathcal{F}^* = \{(s_i, r_i, o_i \rightarrow \hat{o}_i)\}$, where $\hat{o}_i$ represents the target new object that replaces the original $o_i$. Notably, KE should not compromise the general ability of the model (Huang et al., 2024), as these abilities are usually developed from extensive pre-training.

As shown in Fig. 3, during inference, the hidden state $\mathbf{h}^l$ of the prediction at the $l$-th FFN layer of an LLM is computed according to the following recursive form:

$$\mathbf{h}^l = \mathbf{h}^{l-1} + \mathbf{a}^l + \mathbf{m}^l, \quad \mathbf{m}^l = \mathbf{W}_{\text{out}}^l \sigma \left( \mathbf{W}_{\text{in}}^l (\mathcal{N}(\mathbf{h}^{l-1} + \mathbf{a}^l)) \right), \tag{1}$$

where $\mathbf{a}^l$ and $\mathbf{m}^l$ are the outputs of the attention block and FFN layer, respectively, $\mathbf{W}_{\text{in}}^l$ and $\mathbf{W}_{\text{out}}^l$ represent the weight matrices of the $l$-th FFN layer, respectively, $\mathcal{N}(\cdot)$ represents the layer normalization, and $\sigma(\cdot)$ denotes the activation function. We follow previous research (Meng et al., 2022; 2023) by modeling the FFN layer as operating linear key-value memories as follows:

$$\mathbf{k}^l = \sigma \left( \mathbf{W}_{\text{in}}^l (\mathcal{N}(\mathbf{h}^{l-1} + \mathbf{a}^l)) \right), \quad \mathbf{v}^l = \mathbf{W}_{\text{out}}^l \mathbf{k}^l. \tag{2}$$

Then, the textual knowledge $(s, r, o)$ can be linked to the parametric knowledge of the LLM through the activation derived from the inference process. In this context, the key vector $\mathbf{k}$ can be interpreted as encoding the query $(s, r)$, while the object $o$ is subsequently decoded by the model based on the value vector $\mathbf{v}$ associated with the key (Geva et al., 2021).

L&E methods aim at adjusting the activation value $\mathbf{v}$ on new facts by modifying the parameter $\mathbf{W}_{\text{out}}$. In this paradigm, a learnable perturbation is added to the activation $\mathbf{v}^l$ in the specified layer $l$ using supervised learning, resulting in a new activation $\hat{\mathbf{v}}^l$ that allows the model to generate new answers $\hat{o}$. Then the perturbation matrix $\Delta^l$ should satisfy

$$(\mathbf{W}_{\text{out}}^l + \Delta^l)\mathbf{k}_i^l = \hat{\mathbf{v}}_i^l \quad \text{and} \quad (\mathbf{W}_{\text{out}}^l + \Delta^l)\mathbf{k}_j^l = \mathbf{v}_j^l \tag{3}$$

where $\hat{\mathbf{v}}_i^l$ corresponds to the new facts and $(\mathbf{k}_j^l, \mathbf{v}_j^l)$ to the sampled knowledge that shall be preserved.

For simplified notation, we denote the parameter to be updated $\mathbf{W}_{\text{out}}^l \in \mathbb{R}^{d_2 \times d_1}$ by $\mathbf{W}$, where $d_1$ and $d_2$ represent the dimensions of the intermediate and output layers of the FFN, respectively. To update a large batch of facts, we obtain the key and value matrix by stacking the vectors:

$$\mathbf{K}_1 = [\mathbf{k}_1, \mathbf{k}_2, \cdots, \mathbf{k}_m] \in \mathbb{R}^{d_1 \times m}, \quad \hat{\mathbf{V}}_1 = [\hat{\mathbf{v}}_1, \hat{\mathbf{v}}_2, \cdots, \hat{\mathbf{v}}_m] \in \mathbb{R}^{d_2 \times m}, \tag{4}$$

where $m$ is the number of edited facts, and we provide details on computing $\mathbf{k}_i$ and $\hat{\mathbf{v}}_i$ in Appendix F. Additionally, we define the residual matrix and residual vectors as

$$\mathbf{R}_1 = \hat{\mathbf{V}}_1 - \mathbf{W}\mathbf{K}_1 \quad \text{and} \quad \mathbf{r}_i = \mathbf{R}_1[:, i]. \tag{5}$$

To preserve the general abilities of the post-edited model, current methods (Meng et al., 2023; Huang et al., 2024; Fang et al., 2025), require the sampling of massive facts from Wikipedia[1] to construct the matrix $\mathbf{K}_0$, which typically consists of 100,000 stacked key vectors.

## 3 RETHINKING LOCATE-AND-EDIT METHODS WITH QUERYING KEY-VALUE DATABASE

In this section, we demonstrate that most L&E methods can be treated as querying a Key-Value (KV) database. We support this argument through both theoretical derivation and experimental validation. Specifically, we update all target facts in a single step in executing MEMIT (Meng et al., 2023) and AlphaEdit (Fang et al., 2025), which have been shown to effectively mitigate the degradation of general capabilities by D4S (Huang et al., 2024). For simplicity, we only discuss the updating over one layer. The detailed derivation of MEMIT and AlphaEdit is provided in Appendix G.

---

[1] https://huggingface.co/datasets/wikimedia/wikipedia

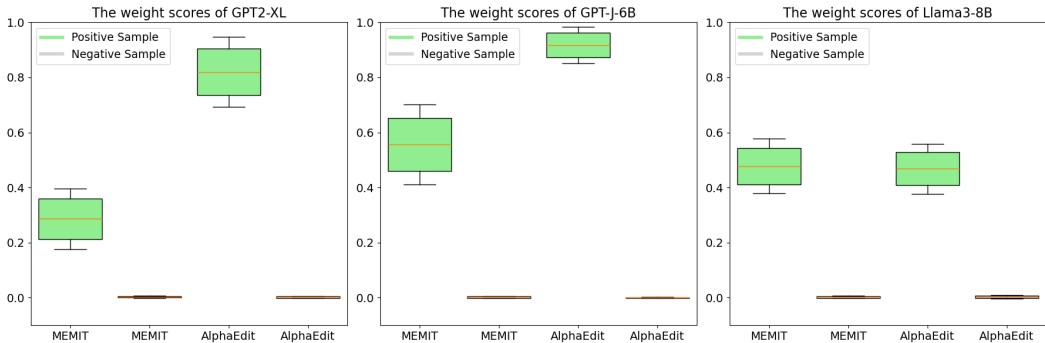

Figure 2: Visualization of weighted scores $\omega = \mathbf{K}_1^T \mathbf{S} \mathbf{k}$ using MEMIT and AlphaEdit for three models. The box-plots are generated from the mean and variance of weight scores, with the center line indicating the mean, boxes showing $\pm 1$ standard deviation, and whiskers $\pm 1.5$.

We conclude that the mechanism of the updating parameter $\Delta_{\text{upd}}$ can be written as

$$(\mathbf{W} + \Delta_{\text{upd}})\mathbf{k} = \mathbf{v} + \mathbf{R}_1 \omega, \quad \omega = \mathbf{K}_1^T \mathbf{S} \mathbf{k} \in \mathbb{R}^{m \times 1}, \tag{6}$$

where $\mathbf{k}$ and $\mathbf{v}$ are the key and value vectors from the original activation, $\mathbf{K}_1$ is key matrix computed from edited facts, and $\mathbf{S}$ is the kernel matrix obtained from specific editing methods.

With self-similarity $\omega = \mathbf{K}_1^T \mathbf{S} \mathbf{k}$ as weighted scores, the result of weighted average $\mathbf{R}_1 \omega$ is integrated into the models to update new knowledge. This means that $\mathbf{k}$ serves as the query to retrieve the residual matrix $\mathbf{R}_1$. Then we discuss the structures of the solutions for MEMIT and AlphaEdit.

The closed-form solution for MEMIT can be derived as follows:

$$\Delta_{\text{upd}}^{\text{MEMIT}} = \mathbf{R}_1 \mathbf{K}_1^T (\mathbf{K}_1 \mathbf{K}_1^T + \beta_1 \mathbf{K}_0 \mathbf{K}_0^T)^{-1}. \tag{7}$$

Let $\mathbf{S}_1 = (\mathbf{K}_1 \mathbf{K}_1^T + \beta_1 \mathbf{K}_0 \mathbf{K}_0^T)^{-1}$, then the update can be expressed as $\Delta_{\text{upd}}^{\text{MEMIT}} = \mathbf{R}_1 \mathbf{K}_1^T \mathbf{S}_1$.

Similarly, AlphaEdit utilize the null space projection matrix $\mathbf{P}$ as the hard constraints of general knowledge. The closed-form solution for AlphaEdit has a similar structure as following:

$$\Delta_{\text{upd}}^{\text{AlphaEdit}} = \mathbf{R}_1 \mathbf{K}_1^T \mathbf{P}^T (\mathbf{P} \mathbf{K}_1 \mathbf{K}_1^T \mathbf{P}^T + \beta_2 \mathbf{I})^{-1} \mathbf{P}. \tag{8}$$

Let $\mathbf{S}_2 = \mathbf{P}^T (\mathbf{P} \mathbf{K}_1 \mathbf{K}_1^T \mathbf{P}^T + \beta_2 \mathbf{I})^{-1} \mathbf{P}$, this can be rewritten as $\Delta_{\text{upd}}^{\text{AlphaEdit}} = \mathbf{R}_1 \mathbf{K}_1^T \mathbf{S}_2$, which is also the weighted average over $\mathbf{R}_1$, with $\mathbf{K}_1^T \mathbf{S}_2 \mathbf{k}$ representing the weighted scores $\omega$. In particular, AlphaEdit returns $\mathbf{0}$ when $\mathbf{k}$ is from the null space of general knowledge $\mathbf{K}_0$. Consequently, $\mathbf{S} \mathbf{k} = \mathbf{P}^T (\mathbf{P} \mathbf{K}_1 \mathbf{K}_1^T \mathbf{P}^T + \beta_2 \mathbf{I})^{-1} (\mathbf{P} \mathbf{k}) = \mathbf{0}$, which effectively preserves the general abilities.

We empirically visualize the weighted scores $\omega = \mathbf{K}_1^T \mathbf{S} \mathbf{k}$ for three post-edited models using MEMIT and AlphaEdit during their inference on new facts. Specifically, we perform KE on $1,000$ facts from the CounterFact dataset and evaluate the post-edited models on these edited facts. When testing the $i$-th edited knowledge, we refer to the $i$-th component of $\omega_i$ as the positive sample, while the remaining components are considered negative samples.

As shown in Fig. 2, the weighted scores labeled as negative samples are close to $0$, while the weighted scores of positive samples are significantly higher, with those for AlphaEdit approaching $1$. Additional results provided in the Appendix J.9 demonstrate that methods typically yield $\omega = 0$ when conducting inference on unmodified facts. The empirical results of these optimized KV databases reveal an important finding: **they return the residual vector corresponding to the edited fact, while returning the zero vector for unrelated questions**.

## 4 METHOD

To overcome the limited capacity of linear system, we construct the neural KV database equipped with non-linear retrieval function, as previous editing methods were primarily optimized for retrieving the most relevant residual vector. As shown in Fig. 3, the proposed NeuralDB editing framework serves as a plug-and-play module for efficient massive knowledge editing.

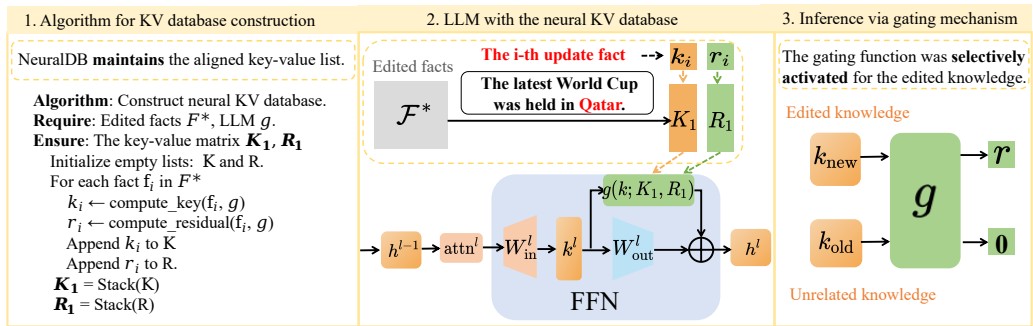

Figure 3: **Overview of NeuralDB editing framework.** (1) The key $\mathbf{K}_1$ and residual matrix $\mathbf{R}_1$, defined in Eq. 4 and Eq. 5, are computed to construct the neural KV database. (2) Our proposed module is designed for efficient, plug-and-play editing. (3) During inference, our non-linear gated function $g(\cdot; \mathbf{K}_1, \mathbf{R}_1)$ only returns the most matched residual $\mathbf{r}_j$ when post-edited models infer one edited fact and involve key vectors $\mathbf{k}_{\text{edited}}$. The function $g(\cdot; \mathbf{K}_1, \mathbf{R}_1)$ reverts zero vector $\mathbf{0}$ when involving the key vector $\mathbf{k}_{\text{pre}}$ of general knowledge.

## 4.1 NEURAL KV DATABASE

Given the target facts $\mathcal{F}^* = \{(s_i, r_i, o_i \rightarrow \hat{o}_i)\}$, we first compute their key and residual matrix to obtain $\mathbf{K}_1$ and $\mathbf{R}_1$. Below, we formally define them as a neural KV database.

**Definition 1** (Neural Key-Value Database). *Given $\mathcal{F}^*$, the constructed neural KV database can be represented as $(\mathbf{K}_1, \mathbf{R}_1)$, where $\mathbf{K}_1 \in \mathbb{R}^{d_1 \times m}$ and $\mathbf{R}_1 \in \mathbb{R}^{d_2 \times m}$ denote the key matrix and residual matrix of the edited facts as defined in Eq. equation 4 and Eq. equation 5. $\mathbf{K}_1$ and $\mathbf{R}_1$ serve as keys and values within the database, with $\mathbf{k}_i = \mathbf{K}_1[:, i]$ being associated with $\mathbf{r}_i = \mathbf{R}_1[:, i]$.*

## 4.2 NONLINEAR GATED RETRIEVAL MODULE

The findings in Section 3 indicate two key requirements in KE: (i) determining whether to use residuals for editing, and (ii) identifying which residual to edit based on the given query. To rigorously implement these demands in the KV database $(\mathbf{K}_1, \mathbf{R}_1)$, we propose the following nonlinear retrieval function, which returns the residual with maximal similarity if it meets the gating condition:

$$g(\mathbf{k}; \mathbf{K}_1, \mathbf{R}_1) = \mathbf{r}_j \cdot \overbrace{\mathbf{1}_{\cos(\mathbf{k}, \mathbf{k}_j) > \gamma}}^{\text{Gate}}, j = \arg\max \cos(\mathbf{k}, \mathbf{k}_i), \tag{9}$$

where $\cos(\cdot, \cdot)$ denotes the cosine similarity, $\mathbf{1}$ represents the indicator function, and $\gamma$ is the parameter controlling the gating mechanism. Although straightforward, cosine similarity proves to be very effective in our key matching experiments in Appendix E. Additionally, its range from the interval $[0, 1]$ provides good interpretability, making it easy to set $\gamma$.

As illustrated in Fig. 3, the nonlinear function is integrated into the target FFN layer as follows:

$$\mathbf{v}^l = \mathbf{W}^l \mathbf{k}^l + g(\mathbf{k}^l; \mathbf{K}_1, \mathbf{R}_1). \tag{10}$$

When post-edited models involve general knowledge, the involved key vector $\mathbf{k}^l$ generally exhibits low similarity to all key vectors within the matrix $\mathbf{K}_1$. Then the workflow of original model remains same since the gating mechanism is not activated. Therefore, the general ability of LLMs can successfully be preserved. In contrast, when post-edited models encounter revised facts, our retrieval function will recall the most closely related residual vector, because of the high similarity between the keys will satisfy the gating condition.

**The ease of deployment**    NeuralDB only requires maintaining the lists of key and value vectors, offering advantages such as convenient addition, deletion, and modification of edited facts. Additionally, our method eliminates the expensive but ineffective process of estimating general knowledge. Compared to current L&E methods, these advantages provide better flexibility and practicality.

**Single layer versus multi-layer editing**    We just implement our module in one single FFN layer. Although previous L&E methods typically employ multi-layer strategies, we find this strategy offers limited performance gains. We provide a related discussion in Appendix I.

**Additional memory usage and computation time** We show that the additional parameters introduced in the NeuralDB module are controllable. Specifically, the space complexity is $O((d_1 + d_2) \times m)$ for the storage of matrices $\mathbf{K_1} \in \mathbb{R}^{d_1 \times m}$ and $\mathbf{R_1} \in \mathbb{R}^{d_2 \times m}$, where $m$ denotes the number of facts. When editing 10,000 facts with Llama 3 8B (Instruct), the additional parameter size amounts to 150M, which constitutes approximately $2.2\%$ of the original model's size. We report the additional running time of in Table 1. The evaluation time for $10,000$ facts in CounterFact dataset increases only by $1.5\%$ compared to the original model. Detailed information regarding additional computation is provided in Appendix H.

Table 1: The ratio of additional time across the number of edits on Llama 3 8B (Instruct).

| Total number of edited facts | 2k | 4k | 6k | 10k | 12k | 16k | 20k |
|---|---|---|---|---|---|---|---|
| **The ratio of additional time** | 0.65% | 1.65% | 1.67% | 1.7% | 2.29% | 3.69% | 5.55% |

## 5 EXPERIMENTS

In this section, we present a comprehensive evaluation of our method for massive KE. For fair comparison, we mostly follow the experimental setups in AlphaEdit (Fang et al., 2025) to benchmark our method. Specifically, we examine whether the edited models can effectively balance mastery of the edited facts with retention of their general capabilities. Detailed implementation of the NeuralDB editing framework is provided in Appendix D. We provide the detailed ablation studies in Appendix I, including $\gamma$ selection, layer selection and multi-layer. Additional results are given in Appendix J, including support for more LLMs, further comparisons with WISE and T-Patcher, results on KnowEdit (Zhang et al., 2024b) and MQuAKE (Zhong et al., 2023), and other supplementary findings.

### 5.1 SET UP

We evaluates the post-edited models after editing all the $T$ facts, where $T$ is the total edited numbers.

**Models and methods** We select three representative LLMs, including GPT-2 XL (Radford et al., 2019), GPT-J (6B) (Wang, 2021), and Llama3 (8B) (Grattafiori et al., 2024). We compare our method with the following KE methods, including Fine-Tune (FT) (Meng et al., 2023), MEND (Mitchell et al., 2022a), ROME (Meng et al., 2022), MEMIT (Meng et al., 2023), SERAC (Mitchell et al., 2022b), GRACE (Hartvigsen et al., 2023a), RECT (Gu et al., 2025), and AlphaEdit (Fang et al., 2025). We provide a detailed introduction on these baselines and models in Appendix B. We provide the results including more LLMs in Section J.10. Additionally, we compare with the WISE (Wang et al., 2025b) in Appendix J.1.

**Datasets for knowledge editing** To evaluate KE methods, we utilize two widely recognized benchmarks: the CounterFact dataset (Meng et al., 2022) and the ZsRE dataset (Levy et al., 2017). Consistent with previous research (Meng et al., 2022; Fang et al., 2025), we employ the following evaluation metrics: *efficacy* (success of edited facts), *generalization* (success of paraphrased facts), *specificity* (success of neighboring facts), *fluency* (generation entropy), and *consistency* (reference score). Detailed explanations of the datasets and metrics are provided in Appendix C.

**Datasets for general ability** We assess the general capabilities of the edited LLMs using the following typical datasets, SciQ (Welbl et al., 2017) (science question answering), MMLU (Hendrycks et al., 2021) (massive multitask language understanding), Commonsense QA (Talmor et al., 2019) (commonsense knowledge understanding), ARC Challenge (Clark et al., 2018) (challenge task requiring reasoning), WSC273 (Kocijan et al., 2019) (coreference resolution), Lambada (Paperno et al., 2016)(predict the endings of text passages). Datasets such as SciQ, MMLU, and Commonsense QA primarily evaluate knowledge-based question answering, focusing on the models' ability to understand and retain factual information. In contrast, the ARC Challenge, WSC273, and Lambada are designed to assess capabilities beyond mere knowledge memory, such as reasoning and text generation. Detailed information regarding these datasets is provided in Appendix C.

Table 2: Comparison of NeuralDB on KE benchmarks. Pre-edited refers to the original models prior to any edits. We evaluated the performance of editing 2,000 facts, with results for FT, MEND, InstructEdit, MELO, and ROME (sourced from Fang et al. (2025)). For 10,000 facts, the results for MEMIT, RECT, AlphaEdit, and NeuralDB are denoted using the arrow ($\rightarrow$) notation. The best results for both 2,000 and 10,000 facts are highlighted in bold, respectively.

| Method | Model | CounterFact | | | | | ZsRE | | |
|---|---|---|---|---|---|---|---|---|---|
| | | Efficacy↑ | Generalization↑ | Specificity↑ | Fluency↑ | Consistency↑ | Efficacy↑ | Generalization↑ | Specificity↑ |
| Pre-edited | | 7.9 | 10.6 | 89.5 | 635.2 | 24.1 | 37.0 | 36.3 | 31.9 |
| FT | LLaMA3 | 83.3 | 67.8 | 46.6 | 233.7 | 8.8 | 30.5 | 30.2 | 15.5 |
| MEND | | 63.2 | 61.2 | 45.4 | 372.2 | 4.2 | 0.9 | 1.1 | 0.5 |
| SERAC | | 71.2 | 61.1 | 66.9 | 615.7 | 20.8 | 67.8 | 34.0 | 22.2 |
| GRACE | | 96.7 | 50.1 | 72.2 | 620.4 | 23.8 | 93.6 | 1.0 | 31.9 |
| ROME | | 64.4 | 61.4 | 49.4 | 449.1 | 3.3 | 2.0 | 1.8 | 0.7 |
| MEMIT | | 63.5→63.4 | 62.8→56.6 | 52.0→50.55 | 466.6→460.4 | 6.5→6.5 | 36.7→0.1 | 32.9→0.1 | 19.1→1.5 |
| RECT | | 64.2→60.0 | 62.5→53.9 | 58.9→51.2 | 502.8→399.1 | 12.9→1.6 | 86.8→0.0 | 82.3→0.0 | 31.9→0.0 |
| AlphaEdit | | 99.1→75.8 | **94.0**→63.1 | 68.6→54.0 | 622.7→417.8 | 32.8→7.0 | 94.4→90.5 | 91.3→85.9 | **32.6**→30.3 |
| NeuralDB | | **99.9→99.2** | 86.6→**85.9** | **88.2→85.6** | 632.7→**631.02** | 32.9→**32.6** | **96.3→95.9** | **92.0→91.0** | 31.9→**31.8** |
| Pre-edited | | 16.2 | 18.6 | 83.1 | 621.8 | 29.7 | 26.3 | 25.8 | 27.4 |
| FT | GPT-J | 92.2 | 72.4 | 43.4 | 297.9 | 6.7 | 72.4 | 68.9 | 19.7 |
| MEND | | 46.2 | 46.2 | 53.9 | 242.4 | 3.9 | 0.7 | 0.7 | 0.5 |
| SERAC | | 82.3 | 58.3 | 69.0 | 615.9 | 28.7 | 92.4 | 38.2 | 25.2 |
| GRACE | | 96.5 | 50.1 | 74.4 | **620.6** | 31.6 | 96.5 | 0.4 | 24.8 |
| ROME | | 57.5 | 54.2 | 52.1 | 589.4 | 3.2 | 56.4 | 54.7 | 9.9 |
| MEMIT | | 98.6→48.8 | 95.4→49.3 | 66.1→51.9 | 557.8→281.5 | 36.5→5.1 | 90.5→0.2 | 84.7→0.1 | **30.9**→0.2 |
| RECT | | 98.8→76.3 | 86.3→70.6 | 74.4→54.9 | 618.1→517.3 | 41.2→25.4 | 96.6→53.5 | 91.5→49.6 | 29.0→21.9 |
| AlphaEdit | | **99.8**→91.6 | **96.3**→79.6 | 76.2→60.3 | 618.5→517.8 | **41.9**→6.9 | 99.7→94.2 | **95.9**→86.1 | 28.8→22.5 |
| NeuralDB | | 99.7→**99.1** | 94.6→**93.2** | **80.0→75.7** | 619.8→**620.0** | 41.4→**41.3** | **99.2→98.2** | 95.9→**95.0** | 27.5→**27.0** |
| Pre-edited | | 22.2 | 24.3 | 78.5 | 626.6 | 31.9 | 22.2 | 31.3 | 24.2 |
| FT | GPT2-XL | 63.6 | 42.2 | 57.1 | 519.4 | 10.6 | 37.1 | 33.3 | 10.4 |
| MEND | | 50.8 | 50.8 | 49.2 | 407.2 | 1.0 | 0.0 | 0.0 | 0.0 |
| SERAC | | 72.3 | 58.2 | 64.1 | 595.4 | 27.4 | 92.2 | 36.6 | 20.7 |
| GRACE | | 98.9 | 50.1 | 72.1 | 620.2 | 28.5 | 94.3 | 1.6 | 27.6 |
| ROME | | 54.6 | 51.2 | 52.7 | 366.1 | 0.7 | 47.5 | 43.6 | 14.3 |
| MEMIT | | 93.0→58.5 | 83.3→55.8 | 58.9→56.1 | 481.8→496.2 | 23.2→8.1 | 74.4→3.5 | 66.9→2.8 | **25.87**→2.07 |
| RECT | | 91.8→86.9 | 79.5→69.5 | 64.0→55.0 | 482.1→517.8 | 20.3→10.9 | 82.6→27.5 | 74.7→25.1 | 24.6→13.1 |
| AlphaEdit | | 99.4→92.2 | 93.8→76.5 | 65.6→56.5 | 584.0→580.9 | 37.9→29.6 | 93.2→57.1 | 83.5→47.5 | 25.3→13.5 |
| NeuralDB | | **99.8→99.1** | **97.2→95.7** | **74.1→70.9** | 621.5→**619.9** | **42.2→41.7** | **96.3→94.6** | **92.8→91.0** | 25.0→**24.2** |

## 5.2 THE EDITING EFFECTIVENESS OF POST-EDITED MODELS

We evaluated the performance after editing 2,000 facts and also included the results of 10,000 facts in parentheses for MEMIT, RECT, AlphaEdit, and NeuralDB. The results are presented in Table 2 and our method demonstrates exceptional efficacy across various scenarios. We present the following three important perspectives to demonstrate our advantages.

**NeuralDB has great generalization.** The rephrased structure presents a significant challenge for the generalization metric, especially for memory-based methods. Although NeuralDB is also classified as a memory-based method, our approach achieves nearly 90% accuracy, whereas SEARC and GRACE only achieve a success rate of almost 50%.

**NeuralDB has few side effects.** Editing methods must be precise and avoid potential side effects. As observed in Table 2, existing methods often exhibit low specificity metrics and their responses lack fluency. Our method effectively mitigates these side effects and achieves near results compared with pre-edited models in terms of both specificity and fluency. Additionally, our method ensures that the generated text maintains coherence with a high degree of consistency.

**NeuralDB has superior scalability.** In particular, when increasing the number of edited facts from 2,000 to 10,000, our method maintains nearly 99% of the results across all metrics and exhibits stable performance. In contrast, existing methods often experience significant degradation, with AlphaEdit preserving only 70% of the results. When editing 10,000 facts, our method demonstrates a comprehensive advantage across all metrics in two datasets.

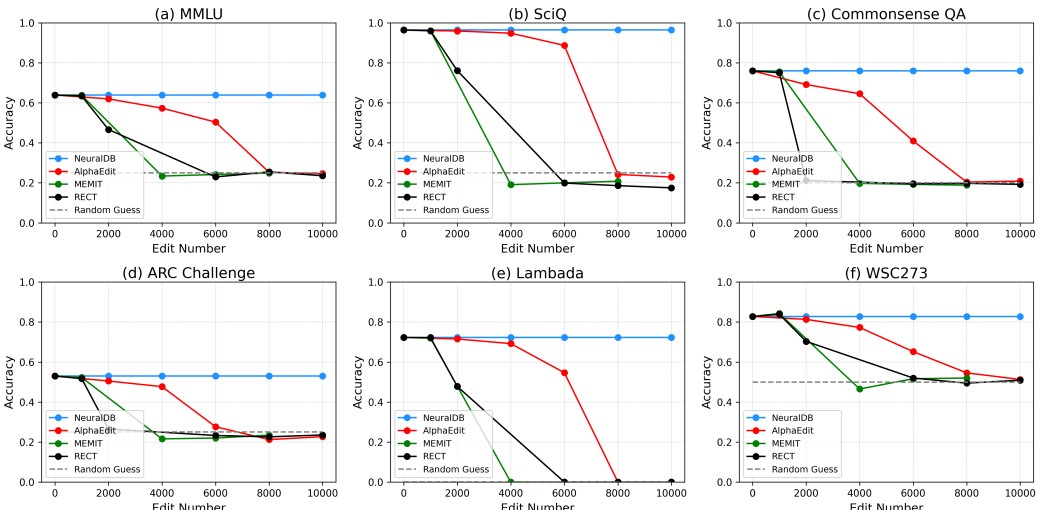

Figure 4: Results over general abilities after massive editing. NeuralDB is evaluated against strong baselines (MEMIT, RECT, and AlphaEdit), with black dashed lines indicating random guessing baselines for multi-choice datasets. The results highlight NeuralDB's advantage: it preserves general capabilities with strong consistency as edit numbers increase significantly.

### 5.3 THE GENERAL ABILITIES OF POST-EDITED MODELS

We assessed post-edited models with various configurations, evaluating their performance across 2,000, 4,000, 6,000, 8,000, and 10,000 facts, as depicted in Fig. 4. The evaluation was conducted on lm-evaluation-harness (Gao et al., 2024). The results show that our method effectively edits a large number of facts without compromising the general abilities of the models across various tasks. In contrast, existing L&E methods struggle with 4,000 facts editing and exhibit a rapid decline in general abilities as the number of edited facts increases. Notably, these baseline methods achieve favorable results on the SciQ dataset, likely due to the dataset's content being well-represented in Wikipedia and thus captured by the sampled $K_0$. However, their performance deteriorates on other tasks, highlighting the limitations of relying on Wikipedia-sampled $K_0$. Our method, which directly incorporates the gated mechanism, offers a more precise and effective approach compared to approximations derived from Wikipedia. For results on more models, please see Appendix J.

### 5.4 SCALING UP THE NUMBER OF EDITED FACTS INTO 100K

We further examine the scalability of the NeuralDB when applied to an extensive volume of knowledge. To obtain a sufficiently large set of facts for this investigation, we utilized the training set of the ZsRE dataset for model editing. The results for the Llama3 8B (Instruct) model are presented in Table 3. These results demonstrate that the performance of NeuralDB remains highly stable as the number of edited facts increases from 10,000 to 100,000 with only marginal degradation observed. In evaluations of the model's general ability, we find that scaling up the number of edited facts to 100,000 did not harm the general ability performance and led to a 0.7% improvement in the benchmark datasets. This underscores the superior scalability of our framework.

Table 3: Editing accuracy and the post-edited model's general performance of NeuralDB on Llama 3 (8B) when editing extremely large sets of facts.

| Number of edits | 0k | 10k | 20k | 30k | 40k | 50k | 60k | 70k | 80k | 90k | 100k |
|---|---|---|---|---|---|---|---|---|---|---|---|
| Efficacy (↑) | 37.0 | 96.9 | 96.6 | 96.6 | 96.4 | 96.1 | 96.0 | 95.9 | 95.8 | 95.6 | 95.5 |
| Generalization (↑) | 36.3 | 91.4 | 91.4 | 91.2 | 91.0 | 90.7 | 90.6 | 90.6 | 90.5 | 90.4 | 90.2 |
| Specificity (↑) | 31.9 | 35.1 | 35.3 | 35.2 | 35.2 | 35.2 | 35.2 | 35.2 | 35.1 | 35.1 | 35.1 |
| MMLU[2] (↑) | 56.2 | 56.2 | 56.2 | 56.2 | 56.2 | 56.2 | 56.2 | 56.9 | 56.9 | 56.9 | 56.9 |

[2]The MMLU results are evaluated by the AlphaEdit project rather than the lm-evaluation-harness. Although they use different metrics, both sets of results reflect the general capabilities of the LLMs.

## 6 RELATED WORK

### 6.1 KNOWLEDGE EDITING THROUGH PARAMETER MODIFICATION

**Locate-and-Edit** The L&E paradigm is derived from casual trace experiments (Meng et al., 2022), suggesting that the factual memory of the Transformer models is primarily associated with the FFN layers (Geva et al., 2021). ROME (Meng et al., 2022) is proposed to edit the factual memory of the models by modifying the parameter of the target FFN layer. MEMIT (Meng et al., 2023) extends ROME to support multi-layer and batch editing versions, allowing the editing of thousands of factual knowledge. To address the challenge of post-edited models losing their general capabilities (Li et al., 2024a; Hsueh et al., 2024), several solutions were proposed, including the dumping of sequential editing caches (Huang et al., 2024), null space projection (Fang et al., 2025), and regularization of the weights (Gu et al., 2024b) and singular values (Ma et al., 2025).

**Hypernetwork** KE (De Cao et al., 2021) trains a lightweight biLSTM-MLP editor to convert a single-sample gradient into a low-rank weight delta. MEND (Mitchell et al., 2021) factorizes two-layer gradients into rank-1 vectors and feeds them through a shared MLP, allowing memory-frugal batch edits. InstructEdit (Zhang et al., 2024a) utilizes a complex prompt template so that gradients self-cluster, enabling diverse OOD tasks. All these approaches (Li et al., 2025b) require additional fine-tuning of the hypernetwork with large datasets, incurring considerable computational overhead.

### 6.2 KNOWLEDGE EDITING WITHOUT PARAMETER MODIFICATION

**External memory module** This line of work starts with SERAC (Mitchell et al., 2022c), which augments a frozen model with explicit retrieval memory. T-Patcher (Huang et al., 2023) injects a sparse "key–value–bias" triplet into the final FFN layer. GRACE (Hartvigsen et al., 2023b) stores erroneous hidden states as discrete keys in a dynamic codebook whose values overwrite selected layers whenever the current activation falls inside an $\epsilon$-ball, enabling millisecond-scale. MELO (Yu et al., 2024) uses a hidden-state–indexed database to activate low-rank, per-edit adapter blocks on demand. Although NeuralDB sharesa similar framework with memory-based, the main difference is using the FFN activation of the subject as the key, which provides stronger generalization and scalability. In Appendix E, we provide a detailed discussion and key matching experiments compared to GRACE (Hartvigsen et al., 2023a) to explain our advantages. MEMOIR (Wang et al., 2025a) achieves strong lifelong editing by training sparse masks over side parameters and routing new queries via sparse activation-pattern matching; in contrast, our approach targets *large-batch* editing at scale with an explicit, in-layer KV module. RASE (Han et al., 2023) enhances T-Patcher and ROME by integrating fact retrieval; unlike our in-model retrieval from hidden states, RASE uses sentence-embedding models for *external* retrieval before applying editing.

**Prompt-based approaches** Recent studies utilize prompt engineering to facilitate efficient KE. For example, MemPrompt (Madaan et al., 2022) and IKE (Zheng et al., 2023b) embed updated knowledge into prompts to leverage in-context learning. For multi-hop QA tasks, MQUAKE (Zhong et al., 2023) and RippleEdits (Cohen et al., 2024) introduces a benchmark to evaluate multi-hop KE performance. MeLLo (Zhong et al., 2023) stores edited facts externally and iteratively prompts the model to yield answers consistent with the updates. PokeMQA (Gu et al., 2024a) improves retrieval and answer accuracy by decomposing multi-hop questions via prompts.COMPKE (Cheng et al., 2025) further extends multi-hop knowledge editing to complex questions involving logical operations (Yin et al.). These settings are closely related to neural graph databases (Bai et al., 2025; Fei et al., 2025) and require complex query answering over knowledge graphs. RAE (Shi et al., 2024) retrieves refined facts and enhances the language model through in-context learning using a knowledge graph. To address multilingual KE, ReMaKE (Wang et al., 2024b) integrates newly retrieved multilingual knowledge into prompts.

## 7 CONCLUSION

In this paper, we introduce NeuralDB, a scalable knowledge editing framework designed to construct a neural KV database from edited facts and integrate it into the target FFN layer within LLMs using a non-linear gated function. This integration ensures that the general capabilities of LLMs

are preserved. The neural database is designed to be easily maintained, facilitating efficient addition and modification of edited facts within the models. We conducted comprehensive experiments across various LLMs to validate the effectiveness of our framework. Our results on the ZsRE and CounterFact datasets, utilizing GPT2-XL, GPT-J (6B), and Llama-3 (8B), demonstrate that NeuralDB editing can effectively modify hundreds of thousands of facts without degrading the quality of generated text. Additionally, our findings from six generic text understanding and generation tasks confirm that our method preserves the general abilities of LLMs unrelated to the target edited facts. These results highlight the robustness and scalability of NeuralDB editing, positioning it as a valuable tool for enhancing the adaptability and accuracy of LLMs in diverse applications.

ACKNOWLEDGMENTS

Jingzhao Zhang, Jiazheng Li and Jing Xu were supported by the Shanghai Qi Zhi Institute Innovation Program.

## 8    ETHICS STATEMENT

The wider impacts include positive contributions to society. By improving the efficiency of knowledge updates, models can adapt more quickly to changing environments and information. This not only benefits research and education, but also provides up-to-date information support in critical fields such as healthcare and justice, promoting scientific and timely decision making. Furthermore, the proliferation of knowledge editing will encourage interdisciplinary collaboration, allowing experts from different fields to share and integrate knowledge more effectively to address complex societal issues.

## 9    REPRODUCIBILITY STATEMENT

To ensure reproducibility, we provide the code in the supplementary materials and detailed implementation information in Appendix D. In the README.md file included with the code, we present a step-by-step guide for reproducing our results, which covers loading datasets, preparing the environment, executing the methods, and evaluating the outcomes.

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

## A    THE USE OF LARGE LANGUAGE MODELS

The LLMs were used solely to refine the writing and check for potential typos. No other aspects were involved.

## B    BASELINES

In this section, we present the six baseline methods evaluated in our work. For each method, we adopt the default hyperparameter settings provided in the official code of the corresponding papers.

- **Fine-Tune (FT)** (Meng et al., 2023) is a fine-tuning method that updates the FFN of a transformer layer to incorporate new factual knowledge. The target layer is selected on the basis of its relevance to the knowledge being edited. FT operates by maximizing the likelihood of the target output using the standard next-token prediction loss.

- **MEND** (Mitchell et al., 2022a) introduces a hypernetwork that maps fine-tuning gradients into efficient weight updates for a pre-trained model. By applying low-rank decomposition to the gradients, it reduces parameter complexity and enables lightweight, localized edits without full model retraining.

- **ROME** (Meng et al., 2022) performs KE by interpreting the FFN in a transformer layer as a linear associative memory. It derives key-value pairs from internal activations and computes a weight update that ensures the edited layer produces the desired hidden representation. A rank-one modification is then applied to the FFN weights, aligning the model's internal representations with the new factual knowledge.

- **MEMIT** (Meng et al., 2023) extends the ROME framework to support simultaneous editing of a large number of factual knowledge items. It models the updates as a joint optimization over key-value pairs and applies rank-one modifications to the FFNs. To prevent interference between edits, MEMIT distributes the updates in a top-down manner across critical FFN layers, achieving efficient, scalable, and stable insertion of new factual knowledge.

- **RECT** (Gu et al., 2025) reduces the degradation of general capabilities caused by KE. It regularizes weight updates by constraining their magnitude and selectively updates only the top-$k\%$ of parameters with the largest changes during fine-tuning. This reduces overfitting and helps preserve the model's reasoning and question-answering abilities, while still achieving effective factual edits.

- **AlphaEdit** (Fang et al., 2025) introduces a null-space projection mechanism to preserve existing knowledge during editing. It projects the update direction onto the null space of prior knowledge and then applies the projected perturbation to model parameters. This approach reduces interference with previously edited facts and enables effective integration of new information.

## C  DATASETS AND METRICS

In this section, we describe the datasets and evaluation metrics employed in our experiments.

### C.1  DATASETS

We evaluate our methods on two types of datasets: CounterFact and ZsRE for assessing KE, and six benchmarks including SciQ, MMLU, CommonsenseQA, ARC Challenge, WSC273, and LAMBADA for evaluating the general capabilities of post-edited models.

- **Counterfact** (Meng et al., 2022) is a challenging benchmark that focuses on editing incorrect factual statements in language models. Each instance includes a subject and an incorrect attribute to be updated. To assess edit locality, it provides contrastive prompts involving related but distinct entities, ensuring changes do not affect nearby facts. Additionally, the dataset includes paraphrased and semantically equivalent prompts to evaluate the generalization, fluency, and consistency of the edited model responses.

- **ZsRE** (Levy et al., 2017) is a question-answering dataset commonly used in knowledge editing evaluation. Each example includes a subject, a target answer to be edited, paraphrased questions for testing generalization, and unrelated questions for evaluating locality. The dataset also features human-written question variants, enabling assessment of model robustness to semantically equivalent inputs.

- **SciQ** (Welbl et al., 2017) is a multiple-choice science QA dataset covering topics such as physics, chemistry, and biology. It is used to evaluate a model's ability to recall factual scientific knowledge.

- **MMLU** (Hendrycks et al., 2021) is a multitask benchmark containing questions from 57 academic and professional disciplines. It assesses factual knowledge and generalization across diverse domains in zero- and few-shot settings.

- **CommonsenseQA** (Talmor et al., 2019) is a multiple-choice QA dataset that evaluates a model's ability to reason over everyday commonsense. It focuses on applying implicit world knowledge to select the correct answer among distractors.

- **ARC Challenge** (Clark et al., 2018) is a science QA dataset designed to require reasoning beyond simple retrieval. It contains complex grade-school level questions that challenge a model's problem-solving abilities.

- **WSC273** (Kocijan et al., 2019) is a coreference resolution benchmark derived from the Winograd Schema Challenge. It is designed to test whether a model can correctly identify what ambiguous pronouns refer to, based on context and commonsense reasoning.

- **Lambada** (Paperno et al., 2016) is a word prediction benchmark composed of narrative passages where the final word can only be inferred from the entire context. It is designed to evaluate a model's ability to capture long-range dependencies and maintain discourse-level coherence.

## C.2 METRICS

Here, we introduce the evaluation metrics for the CounterFact and ZsRE datasets, which are selected based on previous works (Meng et al., 2022; Fang et al., 2025).

### C.2.1 COUNTERFACT METRICS

Given a language model $f$, a query $(s_i, r_i)$, an edited object $\hat{o}_i$, and the original object $o_i$, the evaluation metrics for CounterFact are defined as follows.

- **Efficacy (success of edited facts):** The proportion of instances in which the model prefers the edited object $\hat{o}_i$ over the original object $o_i$ when prompted with $(s_i, r_i)$:

$$\mathbb{E}_i \left[ \mathbb{P}_f \left[ \hat{o}_i \mid (s_i, r_i) \right] > \mathbb{P}_f \left[ o_i \mid (s_i, r_i) \right] \right]. \tag{11}$$

- **Generalization (success of paraphrased facts):** The proportion of paraphrased prompts $F_i(s_i, r_i)$, representing rephrasings of the original query $(s_i, r_i)$, for which the model assigns higher likelihood to $\hat{o}_i$ than to $o_i$:

$$\mathbb{E}_i \left[ \mathbb{P}_f \left[ \hat{o}_i \mid F_i(s_i, r_i) \right] > \mathbb{P}_f \left[ o_i \mid F_i(s_i, r_i) \right] \right]. \tag{12}$$

- **Specificity (success of neighborhood facts):** The proportion of neighborhood prompts $N_i(s_i, r_i)$, which involve subjects semantically related to but distinct from the original subject $s_i$, for which the model assigns higher likelihood to the correct object $o_i$ than to the edited object $\hat{o}_i$:

$$\mathbb{E}_i \left[ \mathbb{P}_f \left[ \hat{o}_i \mid N_i(s_i, r_i) \right] < \mathbb{P}_f \left[ o_i \mid N_i(s_i, r_i) \right] \right]. \tag{13}$$

- **Fluency (generation entropy):** Measures output repetition based on the entropy of n-gram distributions in model outputs. Specifically, it computes a weighted combination of bigram and trigram entropies, where $g_n(\cdot)$ denotes the n-gram frequency distribution:

$$-\frac{2}{3} \sum_k g_2(k) \log_2 g_2(k) + \frac{4}{3} \sum_k g_3(k) \log_2 g_3(k). \tag{14}$$

- **Consistency (reference score):** Consistency is measured by prompting the model $f$ with a subject $s$ and computing the cosine similarity between the TF-IDF vectors of its generated output and a reference Wikipedia passage about the object $o$.

### C.2.2 ZSRE METRICS

Given a language model $f$, a query $(s_i, r_i)$, an edited object $\hat{o}_i$, and the original object $o_i$, the evaluation metrics for ZsRE are defined as follows:

- **Efficacy (success of edited facts):** Top-1 accuracy on the edited samples, measuring the proportion of cases in which the model ranks the edited object $\hat{o}_i$ as the most likely prediction given the prompt $(s_i, r_i)$:

$$\mathbb{E}_i \left[ \hat{o}_i = \arg\max_o \mathbb{P}_f \left( o \mid (s_i, r_i) \right) \right] \tag{15}$$

- **Generalization (success of paraphrased facts):** Top-1 accuracy on paraphrased prompts $F_i(s_i, r_i)$, which are rephrasings of the original query $(s_i, r_i)$, measuring the proportion of cases in which the model ranks the edited object $\hat{o}_i$ as the most likely prediction for the given rephrased prompt:

$$\mathbb{E}_i \left[ \hat{o}_i = \arg\max_o \mathbb{P}_f \left( o \mid F_i(s_i, r_i) \right) \right] \tag{16}$$

- **Specificity (success of neighborhood facts):** Top-1 accuracy on neighborhood prompts $N_i(s_i, r_i)$, which involve subjects related to but distinct from $s_i$. Specificity reflects whether the model preserves correct predictions on unaffected inputs by still preferring $o_i$ over $\hat{o}_i$:

$$\mathbb{E}_i \left[ o_i = \arg\max_o \mathbb{P}_f \left( o \mid N_i(s_i, r_i) \right) \right] \tag{17}$$

## D  IMPLEMENTATION DETAILS

We provide the details of the implementation of the experiments. To reproduce our methods, a GPU with 40G memory is required. Our framework is built on L&E methods like MEMIT (Meng et al., 2023) and AlphaEdit (Fang et al., 2025). We first sequentially compute the key vector $\mathbf{k}$ and the residual vector $\mathbf{r}$ for the target edited facts. The details of the computation can be found in Appendix F. Then we stack them as the key matrix $\mathbf{K}_1$ and the residual matrix $\mathbf{R}_1$ and construct a neural KV database $(\mathbf{K}_1, \mathbf{R}_1)$. Then we integrate the non-linear retrieval module with the target FFN layer $l^*$. Our module only involves one hyperparameter $\gamma$ to control the gated mechanisms. We provide the details of the setting in the following Table 4. For the setting $\gamma$, we recommend $\gamma \in [0.6, 0.8]$ and provide the detailed ablation study in Appendix I. For the distance function, We use cosine similarity as the distance function because it has a closed range of $[0, 1]$. across all edited facts, thereby eliminating the need for complex key merging. Other distance functions can make it difficult to determine a suitable value for $\gamma$.

Table 4: Hyper-parameters of NeuralDB for various models in the main experiments

|  | GPT2-xl | GPT-J (6B) | Llama 3 Instruct (8B) |
|---|---|---|---|
| Layer found by casual trace | 17 | 17 | 17 |
| Layer $l^*$ used by NeuralDB | 17 | 8 | 7 |
| gating threshold $\gamma$ | 0.65 | 0.65 | 0.65 |

## E  COMPARATIVE ANALYSIS WITH GRACE

Both GRACE and NeuralDB implement a key-value (KV) database structure for knowledge editing, a design prevalent in memory-based methods due to its elegant modeling. Despite this architectural similarity, our empirical results demonstrate that NeuralDB achieves significantly superior editing performance in large-scale knowledge editing scenarios, particularly in generalization metrics. To systematically investigate the source of this advantage, we first identify the core architectural differences (Table 5) and perform key matching experiments (Table 6).

Both methods can be formally represented as KV databases:

$$h^l = \text{Editing}(h^{l-1}) \quad \text{if} \quad (\text{distance}(k, K_{i^*}) < \gamma_{i^*}) \quad \text{where} \quad i^* = \arg\min_i \text{distance}(h^{l-1}, K_i)$$

where distance is one distance function (e.g. L2, cosine similarity) and $\gamma_{i^*}$ is the threshold for distance function.

The critical distinctions lie in two aspects: (1) NeuralDB utilizes the last token of the subject as the anchor token, which provides unique feature representation and ensures scalability, whereas GRACE uses the last token of the question; (2) NeuralDB selects keys based on activations in the

Table 5: Architectural differences between GRACE and NeuralDB (ours). Illustrated editing the fact "The latest World Cup was held in Qatar."

| Component | NeuralDB | GRACE |
|---|---|---|
| Anchor Token Position | Last token of subject (e.g., **Cup**) | Last token of question (e.g., **is**) |
| Key Selection Mechanism | Activation in FFN layer | Hidden state of given layer |

Table 6: Key matching performance comparison under 100 facts

| Method | Rewrite Matched | Rewrite Unmatched | Rephrased Matched | Rephrased Unmatched | Neighborhood Unmatched |
|---|---|---|---|---|---|
| NeuralDB (Cosine ↑) | 1.00 | 0.18 | 0.84 | 0.19 | 0.18 |
| GRACE (Cosine ↑) | 1.00 | 0.25 | 0.44 | 0.44 | 0.45 |
| GRACE (L2 ↓) | 1.00 | 3968 | 781 | 768 | 948 |

FFN layer, which have been empirically verified as crucial for factual knowledge memorization and provide better generalization capabilities compared to GRACE's hidden state approach.

To empirically validate why our KV structure outperforms GRACE's, we conducted key matching experiments. The results (Table 6) demonstrate that NeuralDB achieves superior key discrimination:

- For **NeuralDB**: Rewrite Matched (1.00) ≫ Rewrite Unmatched (0.18) and Rephrased Matched (0.84) ≫ Rephrased Unmatched (0.19), indicating effective key discrimination even for rephrased queries.
- For **GRACE**: Rephrased Matched (0.44) ≈ Rephrased Unmatched (0.44) in cosine similarity, and even Rephrased Matched (781) > Rephrased Unmatched (768) in L2 distance, revealing its inability to distinguish matched facts for rephrased queries.

The results show that NeuralDB can easily distinguish editing keys from unrelated keys, while GRACE struggles in this regard. This empirical evidence directly explains the generalization performance gap observed in Table 2 (NeuralDB: 86.60 vs. GRACE: 58.58). This confirms that NeuralDB's key discrimination mechanism offers significantly better generalization capabilities compared to GRACE and other memory-based methods.

## F    COMPUTATION OF KEY VECTOR $\mathbf{k}_i$ AND VALUE VECTOR $\mathbf{r}_i$

We follow previous locating-and-editing methods (Meng et al., 2022; 2023; Fang et al., 2025) to derive the key vector and residual vector from the given edited fact $(s_i, r_i, o_i \rightarrow \hat{o}_i)$. Let $l^*$ denote the FFN layer to be updated.

For the key vector $\mathbf{k}_i$, we retrieve the specified activation from LLM inferring the prompt. We denote $\mathbf{k}^{l^*}(\mathbf{x})$ as the key activation of the prompt $\mathbf{x}$ in layer $l^*$. Then the target key vector are computed by the following average over random prefix $\mathbf{x}_j$:

$$\mathbf{k}_i = \frac{1}{N} \sum_{j=1}^{N} \mathbf{k}^{l^*}(\mathbf{x}_j + \mathbf{s}_i), \tag{18}$$

where $\mathbf{s}_i$ is the subject of edited fact. The $\mathbf{x}_j$ is the prefix randomly generated by the language model $f$ to improve the robustness of the expressive ability of $\mathbf{k}_i$.

For the target vector $\mathbf{r}_i$, we wish to find some vector to decode the new answer $\hat{o}_i$. We utilize the supervised learning to derive $\mathbf{r}_i = \arg\min_{\mathbf{r}} L(\mathbf{r})$, where the loss object $L(\mathbf{r})$ is defined as following:

$$\frac{1}{N} \sum_{j=1}^{N} \left( -\log \mathbb{P}_{f(\mathbf{h}^{l^*}+=\mathbf{r})}[o^*|x_j + p] + D_{\mathrm{KL}}(\mathbb{P}_{f(\mathbf{h}^{l^*}+=\mathbf{r})}[x|p'] \| \mathbb{P}_f[x|p']) \right). \tag{19}$$

$p$ is the factual prompt while $p'$ is its variant ( the form of "subject is a"). $f(\mathbf{h}^{l^*}+ = \mathbf{r})$ indicates substituting the output of the $i$-th MLP with an additional learnable parameter $\mathbf{r}$. This optimization also uses the random prefix text $x_j$ to enhance the robustness.

## G    DERIVATION OF SOLUTION TO MEMIT AND ALPHAEDIT

The objective of MEMIT is the following constrained least squares optimization:

$$\arg\min_{\Delta} \|(\mathbf{W} + \Delta)\mathbf{K}_1 - \hat{\mathbf{V}}_1\|_2^2 + \beta_1\|\Delta\mathbf{K}_0\|_2^2, \tag{20}$$

where the term $\|\Delta\mathbf{K}_0\|_2^2$ ensures that the updated parameters maintain general knowledge. With residual matrices $\mathbf{R}_1 = \hat{\mathbf{V}}_1 - \mathbf{W}\mathbf{K}_1$, the closed-form solution for MEMIT can be derived as follows:

$$\Delta_{\text{upd}}^{\text{MEMIT}} = \mathbf{R}_1\mathbf{K}_1^T(\mathbf{K}_1\mathbf{K}_1^T + \beta_1\mathbf{K}_0\mathbf{K}_0^T)^{-1}. \tag{21}$$

Let $\mathbf{S}_1 = (\mathbf{K}_1\mathbf{K}_1^T + \beta_1\mathbf{K}_0\mathbf{K}_0^T)^{-1}$, then the update can be expressed as $\Delta_{\text{upd}}^{\text{MEMIT}} = \mathbf{R}_1\mathbf{K}_1^T\mathbf{S}_1$.

Using SVD decomposition $\mathbf{U}, \mathbf{S}, \mathbf{U}^T = \text{SVD}(\mathbf{K}_0^T\mathbf{K}_0)$, the null space can be obtained by the sub matrix $\mathbf{N} = \mathbf{U}[:, \mathbf{S} < \epsilon]$ by removing the eigenvectors corresponding to non-zero eigenvalues, where $\epsilon$ typically set as $10^{-2}$. Let $\mathbf{P} = \mathbf{N}\mathbf{N}^T$, where $\mathbf{P}\mathbf{K}_0 = \mathbf{N}(\mathbf{N}^T\mathbf{K}_0) \approx \mathbf{0}$.

Similarly, we discuss AlphaEdit, which leverages the null space projection to convert soft constraints for general knowledge into hard constraints. The null space projection matrix $\mathbf{P}$ such that $\mathbf{P}\mathbf{K}_0 = \mathbf{0}$ is computed using SVD decomposition over $\mathbf{K}_0^T\mathbf{K}_0$. Then, AlphaEdit constructs $\Delta = \delta\mathbf{P}$ such that $\|\Delta\mathbf{K}_0\|$ approaches zeros and solves the following least squares problem with L2 Norm:

$$\arg\min_{\delta} \|(\mathbf{W} + \delta\mathbf{P})\mathbf{K}_1 - \hat{\mathbf{V}}_1\|_2^2 + \beta_2\|\delta\mathbf{P}\|_2^2. \tag{22}$$

## H    ADDITIONAL MEMORY USAGE AND COMPUTATION TIME

In this section, we provide a detailed discussion of additional resources of our new module.

**Memory usage**    We cache the key matrix $\mathbf{K}_1$ and the residual matrix $\mathbf{R}_1$ and construct the new module, which totally take $(d_1 + d_2) \times m$ parameters with $m$ denoting the number of edited facts. For Llama 3 8B model with $d_1 = 14,336$, $d_2 = 4,096$, the memory of 10,000 facts is about 150 million parameters. Compared to the total 8B parameters, the additional memory for 1M facts is only $2.2\%$. Additionally, our memory grows linearly with the facts and is easily scaled to more facts.

**Computation time**    We report the average evaluation time for three models and two datasets in Table 7. The results show that the averaged time has only a slight improvement compared with the methods without an additional module.

Table 7: The averaged time of evaluation post-edited models on CounterFact and ZsRE

| Model | Llama3 | | | GPTJ-6B | | |
|---|---|---|---|---|---|---|
| Method | MEMIT | AlphaEdit | NeuralDB | MEMIT | AlphaEdit | NeuralDB |
| CounterFact | 4.12 | 4.11 | 4.18 | 3.81 | 3.76 | 3.90 |
| ZsRE | 0.22 | 0.22 | 0.22 | 0.16 | 0.16 | 0.17 |

## I    ABLATION STUDY

### I.1    LAYER SETTING: SINGLE-LAYER OR MULTI-LAYER?

We provide the pseudo code of the new multi-layer and the old multi-layer strategies in Algorithm 2 and Algorithm 1, respectively. New multi-layer strategy assigns parts of facts to each layer and

---

**Algorithm 1** Old multi-layer method

---

**Require:** Input Transformer model $f$, target layers list $L = [l_1, \cdots, l_n]$, request facts $\mathcal{F}$, Function COMPUTE_KEY to compute the keys of facts at layer $l$, COMPUTE_RESIDUAL compute the residual of facts at layer $l$ .
1: $R \leftarrow$ COMPUTE_RESIDUAL$(f, \mathcal{F}, l_n)$
2: **for** $l$ in $L$ **do**
3:     $K_i \leftarrow$ COMPUTE_KEY$(f, \mathcal{F}, l)$
4:     $R_i \leftarrow$ R$/(l_n - l + 1)$
5:     Perform KE at layer $l$ with $(K_i, R_i)$
6: **end for**

---

**Algorithm 2** New multi-layer method

---

**Require:** Input Transformer model $f$, target layers list $L = [l_1, \cdots, l_n]$, request facts $\mathcal{F}$, Function COMPUTE_KEY to compute the keys of facts at layer $l$, COMPUTE_RESIDUAL compute the residual of facts at layer $l$ .
1: **for** $l$ in $L$ **do**
2:     $K_i \leftarrow$ COMPUTE_KEY$(f, \mathcal{F}, l)$
3:     $R_i \leftarrow$ COMPUTE_RESIDUAL$(f, \mathcal{F}, l)$
4:     Perform KE at layer $l$ with $(K_i, R_i)$
5: **end for**

---

update the key-value pair for these layers separately. The edited layers by new multi-layer strategy will disturb each other, resulting in poor performance. The compassion of these two algorithms on two different pre-trained models in Table 8. The experimental results indicate that the new multilayer method can greatly scale the number of editable facts, albeit with some loss in performance, while the old multilayer method, though achieving better editing accuracy, requires substantially more storage.

Furthermore, to ensure fairness, we provide comparisons with the single-layer versions of AlphaEdit and MEMIT in Table 9. The results clearly demonstrate our improvements and further highlight our advantages.

## I.2 LAYER SELECTION

For the $L = 7$ in Llama3, we provide the results of additional 8 and 9 in Table 10. Although layer 8 is determined by the causal trace, our results show that its results are not suboptimal.

## I.3 GATING HYPERPARAMETER $\gamma$ SELECTION

We conducted an ablation study to investigate the choice of $\gamma$ and the target layer, with results provided in Table 11. Although all the evaluation metrics are usually monotonically changed as $\gamma$ increases, we observe a consistent trend across three models. $\gamma = 0.65$ provides a good balance across metrics for three models. Importantly, the generalization metric is monotonic increasing, while the specificity metric is monotonic decreasing. Extremely low or high $\gamma$ substantially degrades some metrics. Therefore, we recommend selecting $\gamma$ in $[0.55, 0.65]$, which empirically performs a good performance for all the metrics and three models.

## J ADDITIONAL EXPERIMENTS

## J.1 THE COMPARISON WITH WISE

Table 8: Editing performance under different layer setup

| Model | Layer Setup | Efficacy ↑ | Generalization ↑ | Specificity ↑ | Fluency ↑ |
|---|---|---|---|---|---|
| GPT-J | [8] baseline | 99.08 | 93.48 | 75.52 | 620.53 |
| | [6,7,8] new multi layers | 94.44 | 91.72 | 75.93 | 617.44 |
| | [6,7,8] old multi layers | 99.31 | 93.23 | 76.78 | 616.00 |
| GPT2-XL | [17] baseline | 99.04 | 95.96 | 70.72 | 621.90 |
| | [15,16,17] new multi layers | 94.81 | 92.68 | 70.26 | 618.51 |
| | [15,16,17] old multi layers | 99.08 | 94.01 | 71.33 | 624.48 |

Table 9: The comparison with single-layer versions of MEMIT and AlphaEdit. In each value **a/b**, the first is the result of 2k while the second corresponds to 10k.

| Method | Efficacy Counteract | Generalize Counteract | Specificity Counteract | Fluency Counteract | Consistency Counteract | Efficacy ZsRE | Generalize ZsRE | Specificity ZsRE |
|---|---|---|---|---|---|---|---|---|
| MEMIT | 69 / 72 | 59 / 61 | 52 / 46 | 519 / 522 | 6 / 6 | 68 / 0 | 63 / 0 | 27 / 1 |
| AlphaEdit | 97 / 88 | 83 / 73 | 69 / 53 | 623 / 563 | 32 / 29 | 93 / 88 | 88 / 83 | **32.5** / 31 |
| NeuralDB | **100 / 99** | **87 / 86** | **88 / 86** | **633 / 631** | **33 / 33** | **96 / 96** | **92 / 91** | 32 / **32** |

We provide the comparison with WISE in the Table 12. We implement both WISE and NeuralDB using EasyEditor, a popular and useful code repository for knowledge editing. For fairness, we use the default parameters of WISE provided in EasyEditor (WISE is released through EasyEditor).

WISE achieves consistent results across all efficacy, generalization, and locality metrics. However, our method demonstrates superior performance on large-scale edits, as evidenced by its results with 2,000 and 5,000 facts.

## J.2 THE COMPARISON WITH T-PATCHER

We have included the comparison with T-Patch in Table 13, where NeuralDB demonstrates its advantages across two models and two metrics.

## J.3 THE RESULTS ON MQUAKE

The results on MQuAKE are shown in Table 14. Our method outperforms competing baselines and achieves the best overall performance. Compared with AlphaEdit, it delivers a 36% improvement. These results indicate that complex editing tasks requiring intermediate steps also benefit from our approach.

## J.4 THE RESULTS ON KNOWEDIT

We provide the results on KnowEdit (Wiki Recent) and KnowEdit (Wiki Counterfact), both of which include the Portability metric (See Table 15). Our method achieves the best Portability on Wiki Counterfact and comparable Portability to the strongest baselines on Wiki Recent, indicating that the edited knowledge is effectively utilized in downstream reasoning.

## J.5 QUANTITATIVE AND QUALITATIVE RESULTS FOR DELETE OPERATION

The knowledge updating is linked with the KV database. Thus, we can modify and delete the knowledge based on the modification and deletion of the content of the KV database.

Table 10: Ablation study on Llama 3 (8B)

| Gamma | Layer | E | G | S | F | C |
|-------|-------|------|------|------|-------|------|
| 0.65 | 7 | 99.2 | 85.9 | 85.6 | 631.9 | 32.6 |
| 0.65 | 8 | 99.2 | 79.3 | 85.1 | 631.5 | 33.3 |
| 0.65 | 9 | 99.2 | 77.4 | 84.9 | 631.6 | 32.4 |

Table 11: Unified $\gamma$ ablation across three models. Metrics: E (Efficacy↑), G (Generalization↑), S (Specificity↑), F (Fluency↑).

| Model | Gamma | E | G | S | F |
|-------|-------|-------|-------|-------|--------|
| GPTJ-6B | 0.15 | 84.10 | 78.95 | 56.20 | 611.00 |
| | 0.40 | 97.55 | 97.02 | 63.92 | 600.47 |
| | 0.55 | 99.75 | 98.22 | 72.55 | 620.01 |
| | 0.65 | 99.80 | 97.20 | 74.10 | 621.50 |
| | 0.75 | 99.75 | 93.35 | 75.00 | 621.75 |
| | 0.90 | 98.55 | 57.38 | 76.68 | 621.35 |
| GPT2-XL | 0.15 | 94.85 | 90.30 | 58.58 | 587.88 |
| | 0.40 | 99.70 | 98.38 | 65.66 | 611.20 |
| | 0.55 | 99.65 | 98.10 | 76.62 | 619.14 |
| | 0.65 | 99.80 | 94.60 | 80.00 | 619.80 |
| | 0.75 | 99.45 | 81.42 | 81.34 | 620.38 |
| | 0.90 | 96.70 | 30.00 | 82.57 | 620.30 |
| Llama-3 (8B) | 0.15 | 93.3 | 86.0 | 70.3 | 632.5 |
| | 0.40 | 98.8 | 95.0 | 75.7 | 640.3 |
| | 0.55 | 99.1 | 91.9 | 83.4 | 631.6 |
| | 0.65 | 99.2 | 87.0 | 85.6 | 632.0 |
| | 0.75 | 99.2 | 74.1 | 86.2 | 632.3 |
| | 0.90 | 98.8 | 28.7 | 86.9 | 633.2 |

**Setting.** We first edit 100 facts on GPT2-XL, then delete the keys and values for the even-indexed facts. This yields three models: (i) pre-edited, (ii) post-edited, and (iii) deleted-even (after removing edits for even facts only).

**Quantitative.** As shown in Table 16, performance on even facts for the deleted-even model matches the pre-edited model, while performance on odd facts is similar to the post-edited model. This demonstrates selective reversibility (deletions roll back only the targeted facts) and retention (non-deleted facts remain effective).

**Qualitative.** We also provide concrete edit–delete-even case study, where the 4th question is "**Autonomous University of Madrid, which is located in**" and the corresponding target answer is "**Sweden**". We report the answers of three models in Table 17. The answers to deleted-even model successfully revert to the original answer to the pre-edited model.

## J.6 The comparison between one-shot editing and batched sequential update for MEMIT and AlphaEdit

We follow the standard procedures for each editing method (Fang et al., 2025). For a fair comparison, we also report one-shot results in which MEMIT and AlphaEdit apply all factual edits in one go. As shown in Table 18, their one-shot performance is not better than the results obtained with batched sequential updates. Therefore, reporting results under batched sequential updates is a fair evaluation setting.

Table 12: The comparison with WISE on ZsRE. We provide the results of Llama3 8B and Llama2 7B chat hf.

| Models | Methods | Efficacy | Generalization | Locality | Efficacy | Generalization | Locality |
|--------|---------|----------|----------------|----------|----------|----------------|----------|
| Llama3 8B | WISE | 33.2 | 32.8 | 100 | 25 | 24.7 | 100 |
| | NeuralDB | **71** | **67.4** | 100 | **70.7** | **67.3** | 100 |
| | WISE | 62.8 | 59.8 | 100 | 51.7 | 49.8 | 100 |
| Llama2 7B | NeuralDB | **70.7** | **66.4** | 100 | **69.5** | **65.6** | 100 |

Table 13: Results on ZsRE with 1k edits (T-Patch results taken from Li et al. (2025a)).

| Models | Method | Reliability | Generalization |
|--------|--------|-------------|----------------|
| GPT2-XL | T-Patch | 77.29 | 67.74 |
| GPT2-XL | NeuralDB | 96.03 | 91.62 |
| Llama2 | T-Patch | 62.94 | 48.37 |
| Llama2 | NeuralDB | 99.89 | 91.35 |

## J.7 THE RESULTS OF HIGHLIGHTING THE PREVIOUS EDITED FACTS

Adopting the notation of RASE (Han et al., 2023), the reported result is the stricter Edit Retain Rate (ER) instead of the Success Rate (SR): where $SR = \frac{\sum_{t=0}^{T} I(f_t(x_t)=y_{x_t})}{T}$ and $ER = \frac{\sum_{t=0}^{T} I(f_T(x_t)=y_{x_t})}{T}$. Here, $f_t$ is the model after the t-th edit, and $f_T$ is the final model after all edits.

ER emphasizes whether earlier edits remain effective under the final model, making it stricter than SR. Our strong ER results (Tables 2 and 3) indicate that earlier edits are retained after subsequent edits. To further support this, Table 19 reports performance on the first 10k edited facts after editing 10k, 20k, 30k, 40k, and 50k facts; the stability across these settings highlights the effectiveness of previously edited entries.

## J.8 LM-EVALUATION-HARDNESS

We conducted more experiments on lm-evaluation-hardness using GPT2-XL and GPT-J, see Table 21 for details.

## J.9 WEIGHTED SCORE VISUALIZATION OF PARAPHRASED AND NEIGHBORHOOD FACTS

We further conduct experiments on paraphrased and neighborhood facts to examine the distribution of weighted scores under both MEMIT and AlphaEdit. As shown in Fig. 5, the scores for positive samples in paraphrased facts are consistently higher, while those for negative samples remain close to 0. For neighborhood facts, where all components are considered negative, the scores are likewise consistently close to 0. These results confirm that, during inference, residuals unrelated to the edited facts remain inactive, resulting in near-zero weighted scores.

## J.10 THE RESULTS OF MORE MODELS

We present the results of our method using Qwen2.5 and Llama 3.1 8B Instruct in Table 22. Our methods on these new models also achieve good results.

Table 14: The results of MquaKE on GPTJ-6B models under CoT prompting. We follow the standard evaluation protocol with CoT prompting. After applying all 3,000 edits, we evaluate all 3,000 multi-hop questions.

| Methods | MEMIT | AlphaEdit | NeuralDB |
|---|---|---|---|
| Accuracy | 6.13 | 9.14 | 12.40 |

Table 15: Additional results on KnowEdit (Wiki_Recent) and KnowEdit (Wiki_Counterfact).

| Dataset | Method | Rewrite_acc | Paraphrase_acc | Portability_acc | Entropy |
|---|---|---|---|---|---|
| KnowEdit (Wiki Recent) | MEMIT | 87.96 | 62.94 | 57.19 | 615.10 |
| KnowEdit (Wiki Recent) | AlphaEdit | 90.41 | 71.53 | 59.89 | 606.84 |
| KnowEdit (Wiki Recent) | NeuralDB | 97.90 | 84.23 | 57.11 | 607.87 |
| KnowEdit (Wiki Counterfact) | MEMIT | 69.22 | 48.67 | 45.89 | 615.23 |
| KnowEdit (Wiki Counterfact) | AlphaEdit | 73.05 | 50.81 | 44.11 | 616.13 |
| KnowEdit (WikiCo unterfact) | NeuralDB | 96.58 | 73.15 | 53.98 | 610.82 |

Table 16: Quantitative results before editing, after editing, and after deleting even edits.

| Group | Metric | Pre-edited | Post-edited | Delete even edits |
|---|---|---|---|---|
| Odd-indexed facts | Efficacy | 0.0 | 100.0 | 100.0 |
| Odd-indexed facts | Generalization | 22.0 | 98.0 | 98.0 |
| Even-indexed facts | Efficacy | 0.0 | 100.0 | 0.0 |
| Even-indexed facts | Generalization | 24.0 | 99.0 | 22.0 |

Table 17: Qualitative examples before editing, after editing, and after deleting even-indexed edits.

| State | Output |
|---|---|
| pre-edited | *"the city of Madrid, Spain"* |
| Post-edited | *"Sweden, has been working on a project called "The Future of the Human Body" for the past two years."* |
| deleted-even | *"the city of Madrid, Spain"* |

Table 18: The comparison of one-shot editing and batched sequential updates. The number of edited facts is 10k, and the used model is Llama3 8B Instruct. The batch size of the batched update is 100.

| Model | Method | Efficacy ↑ | Generalization ↑ | Specificity ↑ |
|---|---|---|---|---|
| AlphaEdit | one-shot editing | 88.09 | 82.78 | 31.10 |
| AlphaEdit | batched sequential updates | 90.50 | 85.90 | 30.30 |
| MEMIT | one-shot editing | 0.19 | 0.15 | 0.92 |
| MEMIT | batched sequential updates | 0.10 | 0.10 | 1.50 |

Table 19: Performance on the first 10k edited facts after scaling to different totals.

| Edited facts | 10k | 20k | 30k | 40k | 50k |
|---|---|---|---|---|---|
| Efficacy ↑ | 96.9 | 96.6 | 96.6 | 96.5 | 96.2 |
| Generalization ↑ | 91.4 | 91.4 | 91.2 | 91.3 | 90.8 |
| Specificity ↑ | 35.1 | 35.1 | 35.2 | 35.3 | 35.3 |

Table 20: Performance of each task across different editing budgets (1,000, 2,000, 4,000, 6,000, 8,000, 10,000) under various model–algorithm configurations.

(a) GPT-J (AlphaEdit)

| Task | 1,000 | 2,000 | 4,000 | 6,000 | 8,000 | 10,000 |
|---|---|---|---|---|---|---|
| sciq | 0.9110 | 0.9080 | 0.9060 | 0.8900 | 0.8850 | 0.7410 |
| logiq_a | 0.2151 | 0.2243 | 0.2089 | 0.2181 | 0.2304 | 0.2181 |
| commonsense_qa | 0.2080 | 0.2146 | 0.2048 | 0.1884 | 0.1925 | 0.1785 |
| arc_easy | 0.6658 | 0.6477 | 0.6326 | 0.6010 | 0.5804 | 0.4870 |
| MMLU | 0.2660 | 0.2688 | 0.2622 | 0.2592 | 0.2587 | 0.2535 |
| arc_challenge | 0.3276 | 0.3148 | 0.2901 | 0.2782 | 0.2611 | 0.2261 |
| lambada | 0.6722 | 0.6604 | 0.6057 | 0.5158 | 0.4036 | 0.2203 |
| winogrande | 0.6346 | 0.6227 | 0.6093 | 0.5991 | 0.5730 | 0.5635 |
| wsc273 | 0.8425 | 0.8352 | 0.7985 | 0.7399 | 0.7179 | 0.6264 |

(b) GPT-J (NeuralDB)

| Task | 1,000 | 2,000 | 4,000 | 6,000 | 8,000 | 10,000 |
|---|---|---|---|---|---|---|
| sciq | 0.9160 | 0.9160 | 0.9160 | 0.9160 | 0.9160 | 0.9160 |
| logiq_a | 0.2120 | 0.2120 | 0.2120 | 0.2120 | 0.2120 | 0.2120 |
| commonsense_qa | 0.2080 | 0.2080 | 0.2080 | 0.2080 | 0.2080 | 0.2080 |
| arc_easy | 0.6692 | 0.6692 | 0.6692 | 0.6692 | 0.6692 | 0.6692 |
| MMLU | 0.2695 | 0.2697 | 0.2697 | 0.2695 | 0.2695 | 0.2698 |
| arc_challenge | 0.3396 | 0.3396 | 0.3404 | 0.3404 | 0.3404 | 0.3404 |
| lambada | 0.6829 | 0.6827 | 0.6827 | 0.6821 | 0.6821 | 0.6819 |
| winogrande | 0.6409 | 0.6417 | 0.6417 | 0.6417 | 0.6417 | 0.6409 |
| wsc273 | 0.8242 | 0.8242 | 0.8242 | 0.8242 | 0.8242 | 0.8242 |

(c) GPT-2 XL (AlphaEdit)

| Task | 1,000 | 2,000 | 4,000 | 6,000 | 8,000 | 10,000 |
|---|---|---|---|---|---|---|
| sciq | 0.8250 | 0.8230 | 0.7920 | 0.7440 | 0.6390 | 0.4920 |
| logiq_a | 0.2289 | 0.2273 | 0.2012 | 0.2104 | 0.1951 | 0.1889 |
| commonsense_qa | 0.1908 | 0.1957 | 0.1916 | 0.1974 | 0.2080 | 0.1933 |
| arc_easy | 0.5682 | 0.5484 | 0.4987 | 0.4693 | 0.4066 | 0.3493 |
| MMLU | 0.2618 | 0.2562 | 0.2464 | 0.2312 | 0.2369 | 0.2315 |
| arc_challenge | 0.2423 | 0.2346 | 0.2398 | 0.2108 | 0.1887 | 0.2065 |
| lambada | 0.4881 | 0.4170 | 0.2610 | 0.1467 | 0.0767 | 0.0231 |
| winogrande | 0.5904 | 0.5549 | 0.5564 | 0.5272 | 0.5201 | 0.5067 |
| wsc273 | 0.6520 | 0.6227 | 0.5714 | 0.5861 | 0.5678 | 0.5421 |

(d) GPT-2 XL (NeuralDB)

| Task | 1,000 | 2,000 | 4,000 | 6,000 | 8,000 | 10,000 |
|---|---|---|---|---|---|---|
| sciq | 0.8240 | 0.8290 | 0.8280 | 0.8280 | 0.8280 | 0.8280 |
| logiq_a | 0.2212 | 0.2181 | 0.2181 | 0.2181 | 0.2181 | 0.2197 |
| commonsense_qa | 0.1900 | 0.1933 | 0.1941 | 0.1941 | 0.1941 | 0.1941 |
| arc_easy | 0.5770 | 0.5785 | 0.5848 | 0.5848 | 0.5848 | 0.5848 |
| MMLU | 0.2532 | 0.2545 | 0.2547 | 0.2546 | 0.2543 | 0.2544 |
| arc_challenge | 0.2509 | 0.2509 | 0.2491 | 0.2500 | 0.2500 | 0.2517 |
| lambada | 0.5055 | 0.5053 | 0.5077 | 0.5069 | 0.5065 | 0.5053 |
| winogrande | 0.5770 | 0.5785 | 0.5848 | 0.5848 | 0.5848 | 0.5848 |
| wsc273 | 0.6777 | 0.6667 | 0.6850 | 0.6850 | 0.6850 | 0.6850 |

Table 21: Performance of each task across different editing budgets (1,000, 2,000, 4,000, 6,000, 8,000, 10,000) under various model–algorithm configurations.

(a) GPT-J (AlphaEdit)

| Task | 1,000 | 2,000 | 4,000 | 6,000 | 8,000 | 10,000 |
|---|---|---|---|---|---|---|
| sciq | 0.9110 | 0.9080 | 0.9060 | 0.8900 | 0.8850 | 0.7410 |
| logiq_a | 0.2151 | 0.2243 | 0.2089 | 0.2181 | 0.2304 | 0.2181 |
| commonsense_qa | 0.2080 | 0.2146 | 0.2048 | 0.1884 | 0.1925 | 0.1785 |
| arc_easy | 0.6658 | 0.6477 | 0.6326 | 0.6010 | 0.5804 | 0.4870 |
| MMLU | 0.2660 | 0.2688 | 0.2622 | 0.2592 | 0.2587 | 0.2535 |
| arc_challenge | 0.3276 | 0.3148 | 0.2901 | 0.2782 | 0.2611 | 0.2261 |
| lambada | 0.6722 | 0.6604 | 0.6057 | 0.5158 | 0.4036 | 0.2203 |
| winogrande | 0.6346 | 0.6227 | 0.6093 | 0.5991 | 0.5730 | 0.5635 |
| wsc273 | 0.8425 | 0.8352 | 0.7985 | 0.7399 | 0.7179 | 0.6264 |

(b) GPT-J (NeuralDB)

| Task | 1,000 | 2,000 | 4,000 | 6,000 | 8,000 | 10,000 |
|---|---|---|---|---|---|---|
| sciq | 0.9160 | 0.9160 | 0.9160 | 0.9160 | 0.9160 | 0.9160 |
| logiq_a | 0.2120 | 0.2120 | 0.2120 | 0.2120 | 0.2120 | 0.2120 |
| commonsense_qa | 0.2080 | 0.2080 | 0.2080 | 0.2080 | 0.2080 | 0.2080 |
| arc_easy | 0.6692 | 0.6692 | 0.6692 | 0.6692 | 0.6692 | 0.6692 |
| MMLU | 0.2695 | 0.2697 | 0.2697 | 0.2695 | 0.2695 | 0.2698 |
| arc_challenge | 0.3396 | 0.3396 | 0.3404 | 0.3404 | 0.3404 | 0.3404 |
| lambada | 0.6829 | 0.6827 | 0.6827 | 0.6821 | 0.6821 | 0.6819 |
| winogrande | 0.6409 | 0.6417 | 0.6417 | 0.6417 | 0.6417 | 0.6409 |
| wsc273 | 0.8242 | 0.8242 | 0.8242 | 0.8242 | 0.8242 | 0.8242 |

(c) GPT-2 XL (AlphaEdit)

| Task | 1,000 | 2,000 | 4,000 | 6,000 | 8,000 | 10,000 |
|---|---|---|---|---|---|---|
| sciq | 0.8250 | 0.8230 | 0.7920 | 0.7440 | 0.6390 | 0.4920 |
| logiq_a | 0.2289 | 0.2273 | 0.2012 | 0.2104 | 0.1951 | 0.1889 |
| commonsense_qa | 0.1908 | 0.1957 | 0.1916 | 0.1974 | 0.2080 | 0.1933 |
| arc_easy | 0.5682 | 0.5484 | 0.4987 | 0.4693 | 0.4066 | 0.3493 |
| MMLU | 0.2618 | 0.2562 | 0.2464 | 0.2312 | 0.2369 | 0.2315 |
| arc_challenge | 0.2423 | 0.2346 | 0.2398 | 0.2108 | 0.1887 | 0.2065 |
| lambada | 0.4881 | 0.4170 | 0.2610 | 0.1467 | 0.0767 | 0.0231 |
| winogrande | 0.5904 | 0.5549 | 0.5564 | 0.5272 | 0.5201 | 0.5067 |
| wsc273 | 0.6520 | 0.6227 | 0.5714 | 0.5861 | 0.5678 | 0.5421 |

(d) GPT-2 XL (NeuralDB)

| Task | 1,000 | 2,000 | 4,000 | 6,000 | 8,000 | 10,000 |
|---|---|---|---|---|---|---|
| sciq | 0.8240 | 0.8290 | 0.8280 | 0.8280 | 0.8280 | 0.8280 |
| logiq_a | 0.2212 | 0.2181 | 0.2181 | 0.2181 | 0.2181 | 0.2197 |
| commonsense_qa | 0.1900 | 0.1933 | 0.1941 | 0.1941 | 0.1941 | 0.1941 |
| arc_easy | 0.5770 | 0.5785 | 0.5848 | 0.5848 | 0.5848 | 0.5848 |
| MMLU | 0.2532 | 0.2545 | 0.2547 | 0.2546 | 0.2543 | 0.2544 |
| arc_challenge | 0.2509 | 0.2509 | 0.2491 | 0.2500 | 0.2500 | 0.2517 |
| lambada | 0.5055 | 0.5053 | 0.5077 | 0.5069 | 0.5065 | 0.5053 |
| winogrande | 0.5770 | 0.5785 | 0.5848 | 0.5848 | 0.5848 | 0.5848 |
| wsc273 | 0.6777 | 0.6667 | 0.6850 | 0.6850 | 0.6850 | 0.6850 |

Table 22: The results of NeuralDB on Qwen2.5 and Llama 3.1 8B Instruct. We provide the 2000 and 5000 facts on ZsRE and CounterFact.

| Model | Edit Number | Counterfact | | | | | ZsRE | | |
| Metrics | | Efficacy | Generalization | Specificity | Fluency | Consistency | Efficacy | Generalization | Specificity |
|---|---|---|---|---|---|---|---|---|---|
| Qwen2.5 | 2000 | 99.45 | 90.72 | 84.69 | 625.70 | 33.21 | 99.69 | 91.82 | 38.4 |
| Qwen2.5 | 10000 | 98.99 | 89.85 | 81.97 | 625.33 | 33.08 | 99.15 | 91.79 | 38.27 |
| Llama 3.1 | 2000 | 99.80 | 94.60 | 87.92 | 634.13 | 34.01 | 95.96 | 87.11 | 30.91 |
| Llama 3.1 | 10000 | 99.18 | 93.95 | 85.67 | 634.13 | 33.69 | 95.28 | 88.06 | 30.39 |

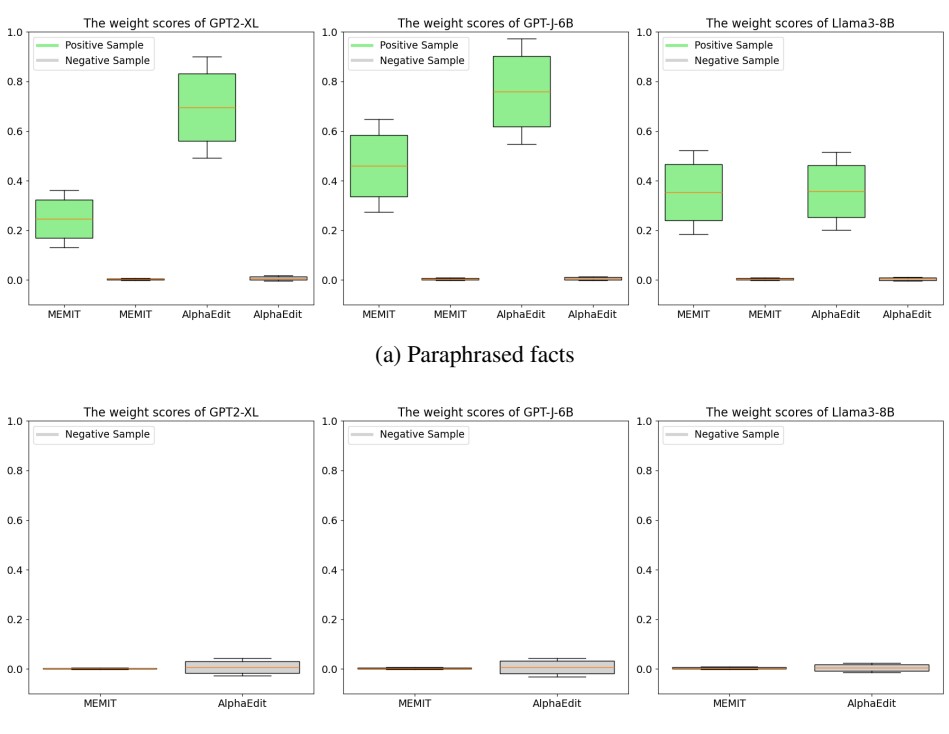

Figure 5: Visualization of weighted scores for paraphrased facts and neighborhood facts, using MEMIT and AlphaEdit across three models. The boxplots are generated from the mean and variance of weight scores, with the center line indicating the mean, boxes showing ±1 standard deviation, and whiskers ±1.5.

