# OpenReview forum: "Scaling Knowledge Editing in LLMs to 100,000 Facts with Neural KV Database"
_ICLR.cc/2026/Conference — ICLR 2026 Poster_

### Official Review · Reviewer_v7LD · 2025-10-21

**Soundness:** 3
**Presentation:** 3
**Contribution:** 3
**Rating:** 4
**Confidence:** 5

**Summary:**

This paper analyzes existing Locate-and-Edit methods and finds that they return the residual vector corresponding to the edited fact, while returning a zero vector for unrelated questions. Based on this observation, the paper proposes **NeuralDB**, which explicitly stores keys and residuals as a KV database. During testing, the most relevant residuals are matched through a **non-linear gated function** and then injected into the hidden state stream.

**Strengths:**

* Viewing the key and residual flows as a KV database is a novel perspective.

* NeuralDB is simple yet highly effective, capable of scaling to a large number of edits while preserving the model’s general capabilities.
* The paper validates the effectiveness of NeuralDB through extensive experiments across multiple models.

**Weaknesses:**

* The paper lacks a comparison with **MEMOIR [1]**, which identifies relevant edits by comparing the sparse activation patterns of new queries with those stored during editing. Both MEMOIR and NeuralDB share a similar **store–identify–inject** paradigm.
* Although NeuralDB demonstrates outstanding performance on the **CounterFact** and **ZsRE** datasets, its effectiveness on **multi-hop reasoning editing tasks** remains unknown, such as **RippleEdit [2]** and **MQuAKE [3]**. If NeuralDB also performs well on these datasets, it would further highlight its applicability.
* NeuralDB requires an additional **KV database**, which increases deployment complexity.

[1] MEMOIR: Lifelong Model Editing with Minimal Overwrite and Informed Retention for LLMs

[2] Evaluating the Ripple Effects of Knowledge Editing in Language Models

[3] MQuAKE: Assessing Knowledge Editing in Language Models via Multi-Hop Questions

**Questions:**

The L&E method requires incorporating the position of the subject token when computing $k$ and $v$. I’m curious whether the $k$ used here is computed in the same way as in L&E. If it is, how is the position of the subject token obtained during testing? If it is not, what exactly is the computation method for $ k $ in this case? Moreover, why is it that even with a different computation method, key matching can still be achieved (i.e., if the last token’s hidden state is used as the key, it should differ from the key at the subject token’s position, and intuitively, the matching success rate should be very low).

---

> ### Author Response · Authors · 2025-11-18
> **The response to Reviewer v7LD**
>
> Dear Reviewer v7LD:
>
> Thank you for your valuable comments and insightful suggestions. We appreciate your recognition of our novel perspective on understanding L&E methods and of the effectiveness and robustness of our approach across multiple models. We wish to invite you to check our new results and discussions, which have been incorporated in the revision.
>
> >Response to weaknesses
>
> >W1. Comparison with MEMOIR.
>
> We appreciate this important suggestion. We were not aware of this new work at the time of our submission.  In the revised manuscript, **we have cited and included a comparison of methodology on page 9.** MEMOIR reports strong results for lifelong (sequential) editing by training sparse masks over side parameters and routing new queries via sparse activation pattern matching. In contrast, our approach targets large-batch editing at scale (up to 100k facts) with an explicit KV module built from in-model activations.  Since the official code of MEMOIR has not been released, an empirical comparison is currently hard.
>
>
> >W2. Additional experiments on the multi-hop reasoning editing task.
>
> Thank you for the valuable suggestion. Due to the limited time, we only present the results on MQuAKE in Table 9. Our method outperforms competing baselines and achieves the best overall performance. Compared with AlphaEdit, it delivers a 36% improvement. These results indicate that complex editing tasks requiring intermediate steps also benefit from our approach.
>
> No methods compared in this work focus on multi-hop knowledge editing. Improving the performance on benchmarks like Mquake can be an important future direction. The results and discussion have also been included in the Appendix in the revised manuscript.
>
> Table 9: The results of MQuaKE on GPTJ-6B models under CoT prompting. We follow the standard evaluation protocol with CoT prompting. After applying all ~3,000 edits, we evaluate all 3,000 multi-hop questions.
>
> | Methods  | MEMIT | AlphaEdit | NeuralDB |
> |---------:|------:|----------:|---------:|
> | Accuracy |  6.13 |      9.14 |    12.40 |
>
>
> >W3. The additional cost of the KV database.
>
> Thank you for sharing the concern. We report memory and latency versus the number of edits in Table 1 of the revised manuscript. For 10k edits, the additional memory and runtime overhead are modest in practice. Because our KV module operates as an in-layer vector store (within the FFN), standard compression and indexing techniques—e.g., low-rank/SVD, quantization, and ANN retrieval—can further reduce both memory and latency.
>
> >Response to Q1: Key matching details
>
> We compute the key k exactly as in L&E. Concretely, for a target layer l, we take the FFN input (pre-activation) at the subject token position as the key vector.  The procedure of key matching is as follows. In practice, subject-based keys most often match the subject tokens in the prompt. Three-step summary:
> - **Build**:  Take (k_i, v_i) pair from the l-layer FFN activation at the subject token position
> - **Integrate**: Insert a small module based on (k_i, v_i) inside the l-layer FFN.
> - **Inference**: This additional module will operate with the hidden states of input tokens at the l-layer FFN. The matched token will be inserted with the value for editing.
>
> **We do not require the subject position at test time.** In key match experiments, we compute cosine similarity between each stored key and all token positions in the evaluated prompt. For key match experiments, a match is counted if any token in the evaluated prompt exceeds the threshold. Although this condition is slack, the failure of key matching can show that the editing would not occur. In general,  the tokens based on the subject will match the subject in the evaluated prompt, while the last position will match the last position.

---

> > ### Comment · Reviewer_v7LD · 2025-11-19
> >
> > Thank you for the detailed response. The key matching requires computing cosine similarity with each token in the prompt, which in the worst case results in quadratic complexity. This means that when the stored key set is very large and the prompt is very long, the matching algorithm would take a significant amount of time. How do you explain or address this issue?

---

> ### Author Response · Authors · 2025-11-19
> **The second response to Reviewer v7LD**
>
> Dear Review v7LD:
>
> Thank you for the kind reply.
>
> Your understanding is correct: key matching computes cosine similarity between prompt tokens and stored keys, where this is inherent to MEMIT, AlphaEdit, and our NeuralDB. Our contribution is to make this **explicit** via a non-linear KV database and to show that we can scale capacity while preserving general abilities.
>
> **Complexity.** The matching cost scales as $O(Lm)$ with prompt length $L$ and number of stored keys $m$. Since the base model’s inference time also grows with $L$, the relative overhead remains stable for a fixed $m$. Moreover, modern LLMs have many layers; our edits perform key matching in **one** FFN layer only.
>
> **Runtime in practice.** The operations of the additional KV database are implemented efficiently in PyTorch. The practical costs in the Counterfact dataset (decoding heavy, requires generating 100 tokens to evaluate fluency)   are shown as modest in Table 2. The editing for 10k edits only introduces 1.7% running time.
>
> **Further acceleration.** Because our KV module operates as a vector database, standard compression and indexing techniques—e.g., low-rank/SVD, quantization, and ANN retrieval—can be applied easily to reduce both memory and latency for massive editing.
>
> Table 2: The ratio of additional latency compared with AlphaEdit
>
> | Total number of Edited facts              | 2k       | 4k       | 6k       | 10k        |  12k        |  16k       | 20k        |
> |----------------------------|:-----:|:-----:|:-----:|:-----:|:-----:|:-----:|:-----:|
> | The ratio of additional time | 0.65% | 1.65% | 1.67% | 1.7% | 2.29% | 3.69% | 5.55% |

---

> > ### Comment · Reviewer_v7LD · 2025-11-19
> >
> > Thank you to the authors for the clarification. However, I do not think that *key matching computes cosine similarity between prompt tokens and stored keys* applies to MEMIT or AlphaEdit, because those methods directly write the new information into the model’s weights. After editing, the inference latency remains the same as in the original model, and their matching mechanism is a mathematically interpretable linear transformation.
> >
> > In contrast, NeuralDB computes cosine similarity, which increases inference latency, and this delay grows with the size of both the key store and the prompt.

---

> > > ### Author Response · Authors · 2025-11-20
> > > **Clarification of the viewpoint on MEMIT and AlphaEdit**
> > >
> > > Dear Review v7LD:
> > >
> > > Thank you for the valuable response.
> > >
> > > We agree with your viewpoint. MEMIT (AlphaEdit) adds essentially no extra inference cost, whereas NeuralDB does. The linear kv database can directly write into the target parameter. We apologize for any confusion.
> > >
> > > Importantly, we want to demonstrate that the added computation is **acceptable and worthwhile** because (1) the overhead is modest, (2) the added module better preserves general abilities, and (3) the additional module can scale to 100k edits. Points (2) and (3) are discussed in the main paper. For (1), **we include complexity analysis, measured runtime, and acceleration options in our response to Reviewer v7LD**. We invite you to review these results and discussions.

---

> > > > ### Comment · Reviewer_v7LD · 2025-11-20
> > > >
> > > > Thank you for your response. I think your paper should clarify in the limitations section that the inference latency of your method increases with the size of the key store and the prompt, and that there may be a risk of incorrect matches. Despite these issues, NeuralDB is indeed a very effective editing method and largely preserves the original capabilities of the model. I have updated my score accordingly.

---

> > > > > ### Author Response · Authors · 2025-11-20
> > > > >
> > > > > Dear Reviewer v7LD,
> > > > >
> > > > > Thank you sincerely for recognizing our work and for your valuable feedback. We will incorporate your suggestions and clearly state NeuralDB’s potential limitations in the revised manuscript. Your insights have been very helpful in improving the paper’s quality.

---

### Official Review · Reviewer_i8Ez · 2025-10-30

**Soundness:** 2
**Presentation:** 3
**Contribution:** 2
**Rating:** 2
**Confidence:** 5

**Summary:**

This paper presents NeuralDB, a scalable knowledge editing framework that can efficiently and robustly integrate up to 100,000 factual edits. NeuralDB reconceptualizes the Locate and Edit (L&E) paradigm as a process of querying a neural key-value (KV) database and introduces a nonlinear gated retrieval module. The proposed approach aims to mitigate catastrophic forgetting of general capabilities and maintain LLM consistency even after large-scale edits. Experiments conducted on three LLM architectures including GPT-2 XL, GPT-J (6B), and Llama-3 Instruct (8B), as well as benchmarks such as ZsRE and CounterFact, demonstrate that as the number of edited facts increases from thousands to 100,000, NeuralDB maintains high editing success rates while preserving performance on general language understanding tasks.

**Strengths:**

1. NeuralDB successfully scales knowledge editing to 100,000 facts, which is an order of magnitude higher than previous methods such as AlphaEdit. As the number of edits increases, NeuralDB maintains both average editing effectiveness and general capability, forming a sharp contrast with the performance degradation observed in baseline methods.
2. The paper provides a coherent reinterpretation of previous Locate and Edit (L&E) approaches as linear key-value (KV) databases, and supports this perspective with mathematical derivations.
3. The NeuralDB approach is simple yet effective, addressing the inherent linearity limitations of existing techniques and enabling robust scalability.

**Weaknesses:**

1. The details before Section 4 are primarily about other work and could be condensed.
2. In the methodology part of Section 4, the approach essentially constructs a plug-and-play knowledge base and uses a gating mechanism to determine when to perform edits. What is the key difference between this idea and "Improving Sequential Model Editing with Fact Retrieval"?
3. In the experimental section, the overall results appear impressive. However, if 10,000 data entries are updated in a plug-and-play manner, the updates primarily rely on the advantages of MEMIT or AlphaEdit. It is recommended to validate the approach on more complex datasets, such as MQuAke, which involves multi-hop editing tasks.
4. The paper mentions T-Patch, but there is no experimental comparison with it.
5. The most critical issue is whether this method continuously modifies the model parameters. For instance, after updating the Nth data entry, do the updates to the 1st to (N-1)th data entries remain effective?

**Questions:**

The comparison in Table 1 is limited to methods that continuously update parameters, which creates an unfair comparison.

---

> ### Author Response · Authors · 2025-11-18
> **The first response to Reviewer i8Ez**
>
> Dear Reviewer i8Ez:
>
> Thank you for your valuable comments and constructive suggestions. We appreciate your acknowledgement of the effectiveness, theoretical contribution, and simple implementation of the proposed method.  Following your feedback, we have updated our manuscript accordingly. We believe the additional discussions and results have effectively improved the quality of the paper.
>
>  We'd like to invite you to check our clarification and the new results below.
>
> >Response to weaknesses
>
> >W5. The effectiveness of previously edited entries
>
> Thank you for the valuable question. We clarify that **our manuscript evaluates the post-edited models after editing all the T facts**, where T is the total number of edited facts. Adopting the notation of RASE[1], the reported result is the stricter **Edit Retain Rate (ER)** instead of the Success Rate (SR): where $SR=\frac{1}{T}\sum_{t=0}^{T}\mathbf{1}(f_t(x_t)=y_{x_t})$ and $ER=\frac{1}{T}\sum_{t=0}^{T}\mathbf{1}(f_T(x_t)=y_{x_t})$. Here, $f_t$  is the model after the t-th edit, and $f_T$ is the final model after all edits.
>
> ER emphasizes whether earlier edits remain effective under the final model, making it stricter than SR. Our strong ER results (Tables 2–3 in the main paper) indicate that earlier edits are retained after subsequent edits.
>
> To further support this, Table 6 reports performance on the first 10k edited facts after editing 10k, 20k, 30k, 40k, and 50k facts; the stability across these settings highlights the effectiveness of previously edited entries.
>
> Table 6: The results of the first 10k edited facts
>
> | Edited facts            | 10k   | 20k   | 30k   | 40k   | 50k   |
> |-------------------|-----:|-----:|-----:|-----:|-----:|
> | Efficacy ↑        | 96.9 | 96.6 | 96.6 | 96.5 | 96.2 |
> | Generalization ↑  | 91.4 | 91.4 | 91.2 | 91.3 | 90.8 |
> | Specificity ↑     | 35.1 | 35.1 | 35.2 | 35.3 | 35.3 |
>
> >W2. The difference with [1] (Improving Sequential Model Editing with Fact Retrieval)
>
> Thank you for sharing this important and relevant work. **We have cited and discussed the differences on page 9 in our revised manuscript.**  RASE improves T-Patch and ROME by incorporating fact retrieval. Our approach likewise maintains a vector key–value store for newly edited knowledge. The key differences are:
>
> 1. Key vector representation. Our method derives keys directly from the LLM’s hidden states at the target layer, whereas RASE employs sentence embeddings.
>
> 2. Where retrieval happens. Our retrieval is in-model (integrated into the FFN layer), while RASE performs external retrieval.
>
> 3. Evaluation protocol. RASE reports both online evaluate-as-you-edit (SR) and post-edit evaluations (ER)  for large-scale editing; our manuscript focuses on post-edit evaluation (ER).
>
> >W4. The comparison with T-Patch
>
> Thank you for the valuable suggestion.  We have included the comparison with T-Patch in Table 7, where  NeuralDB demonstrates its advantages across two models and two metrics.  The results and discussion have been added to the revised manuscript.
>
> Table 7: The results of NeuralDB and T-Patch on the 1k edits in ZsRE. The results of T-Patch are taken from [4]
>
> | Backbone | Method     | Reliability | Generalization |
> |:--------:|:-----------|------------:|---------------:|
> | GPT2-XL  | T-Patch  |       77.29 |          67.74 |
> | GPT2-XL  | NeuralDB   |       **96.03** |          **91.62** |
> | Llama2   | T-Patch  |       62.94 |          48.37 |
> | Llama2   | NeuralDB   |       **99.89** |          **91.35** |

---

> > ### Author Response · Authors · 2025-11-18
> > **The second response to Reviewer i8Ez**
> >
> > >W3. The additional experiments on the multi-hop editing task, MQuAKE[2].
> >
> > Thank you for the valuable suggestion.  The results on MQuAKE are shown in Table 9. Our method outperforms competing baselines and achieves the best overall performance. Compared with AlphaEdit, it delivers a 36% improvement. These results indicate that complex editing tasks requiring intermediate steps also benefit from our approach. The results and discussion have also been included in the Appendix in the revised manuscript.
> >
> > No methods compared in this work focus on multi-hop knowledge editing. Improving the performance on benchmarks like MQuAKE can be an important future direction.
> >
> > Table 9: The results of MquaKE on GPTJ-6B models under CoT prompting. We follow the standard evaluation protocol with CoT prompting. After applying all ~3,000 edits, we evaluate all 3,000 multi-hop questions.
> >
> > | Methods  | MEMIT | AlphaEdit | NeuralDB |
> > |---------:|------:|----------:|---------:|
> > | Accuracy |  6.13 |      9.14 |    12.40 |
> >
> >
> > >W1. The details before Section 4 are primarily about other work and could be condensed.
> >
> > Thank you for the valuable suggestions!  In the revised version, we have condensed the part before Section 4 for better presentation.
> >
> >
> > >Response to Q1: about the fairness comparison with baselines.
> >
> > We follow the standard procedures for each editing method [3]. For a fair comparison, **we also report one-shot results in which MEMIT and AlphaEdit apply all factual edits in one go.** As shown in Table 8, their one-shot performance is not better than the results obtained with batched sequential updates.  Therefore, reporting results under batched sequential updates is a fair evaluation setting.
> >
> > Table 8: The comparison of one-shot editing and batched sequential updates. The number of edited facts is 10k, and the used model is Llama3 8B Instruct.  The batch size of the batched update is 100.
> >
> > | Model     | Method                      | Efficacy ↑ | Generalization ↑ | Specificity ↑ |
> > |:----------|:----------------------------|-----------:|-----------------:|--------------:|
> > | AlphaEdit | one-shot editing            |     88.09  |        82.78     |     31.10     |
> > | AlphaEdit | batched sequential updates  |     90.50  |        85.90     |     30.30     |
> > | MEMIT     | one-shot editing            |      0.19  |         0.15     |      0.92     |
> > | MEMIT     | batched sequential updates  |      0.10  |         0.10     |      1.50     |
> >
> > [1] Improving Sequential Model Editing with Fact Retrieval
> >
> > [2] MQuAKE: Assessing Knowledge Editing in Language Models via Multi-Hop Questions
> >
> > [3] AlphaEdit: Null-Space Constrained Knowledge Editing for Language Models
> >
> > [4] ELDER: Enhancing Lifelong Model Editing with Mixture-of-LoRA

---

> ### Comment · Reviewer_i8Ez · 2025-11-19
> **Batch or sequential editing**
>
> Thank you for your reply.
> Regarding updating T facts, do you mean that NeuralDB can update the results of T facts simultaneously, rather than updating them one by one, such that after updating the i+1 th fact, the update of the i-th fact would disappear?
>
> And for the KV cache, same method used in GRACE, so, if you could update all T facts simultaneously with the cache, that's is a great work, but is you are in sequential, so, please introduce the different to these retrieval or memory augments methods.

---

> > ### Author Response · Authors · 2025-11-19
> > **Response to Batch or sequential editing**
> >
> > Dear Reviewer i8Ez:
> >
> > Thank you for the kind reply.
> >
> > Yes, NeuralDB can update the T facts simultaneously. We provide the pseudo-code for knowledge editing of NeuralDB as following:
> >
> > We first compute the key/value vectors for each fact, stack them, and integrate once. This makes the update order-invariant. After updating  (i+1) fact, the edit to the previous facts does not disappear. Empirically, Table 6 and Tables 2–3 (revised manuscript) confirm this behavior.
> >
> > ```
> >   Algorithm: Knowledge editing in NeuralDB.
> >   Require: T edited facts F, language model h.
> >   Ensure: The edited model \hat{h}
> >   K_1 = torch.stack([compute_key(f_i, h) for f_i in F], dim=0) #shape T \times d_1
> >   R_1 = torch.stack([compute_value(f_i, h) for f_i in F], dim=0) #shape T \times d_2
> >   \hat{h} <— Integrate module g(\cdot; K_1, R_1) into h
> > ```

---

> > > ### Comment · Reviewer_i8Ez · 2025-11-19
> > >
> > > Thank you for explaining this and dispelling my misunderstanding. I am willing to update my score.

---

> > > > ### Author Response · Authors · 2025-11-19
> > > >
> > > > Dear Reviewer i8Ez:
> > > >
> > > > We sincerely appreciate your recognition of our work. We also appreciate all your comments, which have helped us improve the quality of the manuscript.

---

### Official Review · Reviewer_4zaC · 2025-10-31

**Soundness:** 4
**Presentation:** 3
**Contribution:** 3
**Rating:** 8
**Confidence:** 4

**Summary:**

This paper introduces the Neural KV Database (NeuralDB) framework to significantly scale up knowledge editing in Large Language Models (LLMs). The authors re-frame existing Locate-and-Edit (L&E) methods as querying a Key-Value (KV) database, which allows for simultaneous modifications of a massive number of factual knowledge edits. The primary motivation is to overcome the limitations of current L&E methods, which often lead to compromised general abilities and forgetting of previously edited facts when scaling beyond thousands of edits. The proposed NeuralDB claims to scale knowledge editing up to 100,000 facts while maintaining high efficacy and model integrity.

**Strengths:**

1. The paper successfully tackles a critical bottleneck in knowledge editing: scalability. Claiming and demonstrating results up to 10,000 edits (with the promise of 100,000 in the full paper) represents a significant leap forward in the field, far surpassing the typical limits of previous L&E techniques

2. The method directly confronts the major issue of maintaining LLM General Abilities while performing massive knowledge updates, which is vital for practical deployment.

**Weaknesses:**

1. Although the title claims scalability to 100,000 facts, the quantitative experiments only cover up to 10,000 edits, leaving the paper’s core contribution without direct empirical support.

2. Can edited knowledge support reasoning? I suggest adding a Portability metric.

3. How does the framework ensure that the embeddings for 10,000 or 100,000 distinct keys remain sufficiently orthogonal during the retrieval and editing process to prevent mutual interference? If two factual contexts (Keys) are semantically similar, the retrieval mechanism risks activating and modifying non-target KV pairs, leading to subtle, "soft" side effects beyond the targeted edit. The current paper lacks a thorough analysis of this embedding space's decoupling capacity at scale.

**Questions:**

see weaknesses

---

> ### Author Response · Authors · 2025-11-18
> **The response to Reviewer 4zaC**
>
> Dear Reviewer 4zaC:
>
> Thank you for your strong support and encouragement. We are grateful for positive recognition. We also appreciate your constructive suggestions and have included the recommended experiments. The new results and discussions have been added to the revised manuscript, which we believe has greatly improved the work.
>
> Now, we wish to invite you to review the updated results and our responses.
>
>
> >W1. Quantitative experiments on 100,000 facts
>
> Thank you for the question. To support our claim of scalability to 100,000 facts, we report experiments that scale the number of edits from 10k to 100k in Table 3 (Section 5.3).  We present the additional results in Table 4. The results show that both editing metrics and general capability remain consistently stable as we scale to 100k.
>
> Table 4: Scaling methods toward 100k edits. A blank space indicates that no evaluation was conducted.
>
> |  Method         | Edited facts            |    0w   | 1w    | 2w    | 3w    | 4w    | 5w    | 6w    | 7w    | 8w    | 9w    | 10w    |
> |:-----------:|:--------------------:|:-------:|:-------:|:-------:|:-------:|:-------:|:-------:|:-------:|:-------:|:-------:|:-------:|:--------:|
> |NDEdit | Efficacy ↑         | 37.0  | 96.9  | 96.6  | 96.6  | 96.4  | 96.1  | 96.0  | 95.9  | 95.8  | 95.6  | 95.5   |
> |    | Generalization↑    | 36.3  | 91.4  | 91.4  | 91.2  | 91.0  | 90.7  | 90.6  | 90.6  | 90.5  | 90.4  | 90.2   |
> |           | Specificity↑       | 31.9  | 35.1  | 35.3  | 35.2  | 35.2  | 35.2  | 35.2  | 35.2  | 35.1  | 35.1  | 35.1   |
> |           | MMLU  ↑ | 56.2  | 56.2  | 56.2  | 56.2  | 56.2  | 56.2  | 56.2  | 56.9  | 56.91 | 56.9  | 56.9   |
> | AlphaEdit          | Efficacy ↑         | 37.0  | 93.2  |       | 75.3  |       | 22.0  |       |       |       |       |        |
> | | Generalization↑    | 36.3  | 88.3  |       | 66.7  |       | 19.3  |       |       |       |       |        |
> |           | Specificity↑       | 31.9  | 33.8  |       | 23.8  |       | 4.7   |       |       |       |       |        |
> |           | MMLU  ↑ | 56.2  | 50.4  |  | 8.6   |    | 0     |       |       |       |       |        |
>
> >W2. Additional results, including the portability metric (reasoning ability)
>
> Thank you for the insightful suggestion. We now report results on KnowEdit (Wiki_Recent) and KnowEdit (Wiki_Counterfact), both of which include the Portability metric (See Table 5). Our method achieves the best Portability on Wiki_Counterfact and comparable Portability to the strongest baselines on Wiki_Recent, indicating that the edited knowledge is effectively utilized in downstream reasoning.
>
> Table 5: The additional results on Wiki_Recent and Wiki_Counterfact
>
> | Dataset                     | Method     | Rewrite_acc | Paraphrase_acc | Portability_acc | Entropy |
> |----------------------------|------------|-------------:|----------------:|-----------------:|--------:|
> | KnowEdit (Wiki_Recent)     | MEMIT      |       87.96  |          62.94  |            57.19 |  615.10 |
> | KnowEdit (Wiki_Recent)     | AlphaEdit  |       90.41  |          71.53  |            59.89 |  606.84 |
> | KnowEdit (Wiki_Recent)     | NeuralDB   |       97.90  |          84.23  |            57.11 |  607.87 |
> | KnowEdit (Wiki_Counterfact)| MEMIT      |       69.22  |          48.67  |            45.89 |  615.23 |
> | KnowEdit (Wiki_Counterfact)| AlphaEdit  |       73.05  |          50.81  |            44.11 |  616.13 |
> | KnowEdit (Wiki_Counterfact)| NeuralDB   |       96.58  |          73.15  |            53.98 |  610.82 |
>
> >W3. The embedding separability of 100k edits
>
> Thank you for the valuable question on embedding separability. We report results with 100k edits in Table 4 (taken from Table 3 of the main paper). Across this scale, all four metrics remain stable, indicating limited interference among keys as the database grows. In our empirical setting, the unintended activations (side effects) are small.

---

### Official Review · Reviewer_HArh · 2025-11-01

**Soundness:** 4
**Presentation:** 3
**Contribution:** 3
**Rating:** 6
**Confidence:** 4

**Summary:**

This paper tackles a key limitation of existing Locate-and-Edit (L&E) knowledge editing (KE) methods: their failure to scale to thousands of facts without causing catastrophic forgetting and degrading the model's general abilities. NeuralDB2 scales knowledge editing to 100 k facts by replacing implicit linear L&E updates with an explicit gated neural KV memory, curbing catastrophic forgetting and preserving model competence. Finally the paper demonstrates impressive experimental results, showing NeuralDB scales to 10,000 and even 100,000 facts while maintaining high editing success and preserving general model performance.

**Strengths:**

1. **Fresh Perspective with Solid Grounding**: The paper cleverly rethinks existing linear editing methods like MEMIT and AlphaEdit as lookups in a hidden key-value store—backed up by both theory and experiments that show the updates behave almost like one-hot vectors.

2. **Smart, Simple Fix**: Once they spot that sparsity, they swap out the cramped linear system for an explicit neural KV database. A lightweight, cosine-based gate fetches the right “patch” or returns zero if nothing matches—clean and intuitive.

3. **Scales Like a Champ**: NeuralDB keeps its accuracy even after 10 k edits and still protects the model’s general skills, something earlier SOTA methods couldn’t do; they even push the demo to 100 k facts without breaking a sweat.

**Weaknesses:**

**1. On the Sensitivity and Selection of the Gating Hyperparameter $\gamma$**

My primary concern is the selection of the gating threshold $\gamma$, which is the single most critical hyperparameter for preserving general abilities. In the main experiments, the authors use a single value ($\gamma=0.65$) across three different models (GPT2-XL, GPT-J, and Llama-3). This appears to be an unsubstantiated, convenient assumption. This concern is amplified by the ablation study in Appendix I.2 (Table 11), which was *only* conducted on Llama-3. This study reveals that the model's performance is *extremely sensitive* to this parameter. For instance:
Increasing $\gamma$ from *0.65* to *0.75* causes the Generalization (G) score to drop sharply from 85.9 to 74.1. At $\gamma=0.9$, the Generalization ability collapses almost entirely to 28.7.

Given this high sensitivity, how can the authors justify applying $\gamma=0.65$ to the other models without a similar ablation? It is highly probable that the optimal (or even safe) $\gamma$ is model-dependent. This sensitivity implies a very narrow "safety boundary." How can we be confident that unrelated knowledge queries ($k_{old}$) won't accidentally cross this sensitive threshold as the database scales, especially if the chosen $\gamma$ is not truly optimal for that specific model architecture?

**2. On the True Cost of "Scalability" (Memory and Compute)**

The paper's claims of scalability and "controllable" overhead seem to obscure a significant trade-off in memory and compute.

**Memory:** The authors state in Appendix G that editing 10,000 facts for Llama-3-8B adds 150M parameters.This implies that the paper's headline 100,000-fact model carries approximately **1.5B** in additional parameters. For an 8B model, this is an extra 19% in size. This is a very significant cost and can hardly be described as "controllable." This trade-off should be made explicit in the main paper.
**Computation:** More critically, the gated retrieval (Eq. 11)  requires calculating the cosine similarity of the current key $k^l$ against *all $m$ keys* in $K_1$, an $O(m)$ operation for every forward pass. The claim of only a 1.5% evaluation time increase for $m=10,000$ is surprising. Does this latency scale linearly with $m$? If so, the $m=100,000$ model would be substantially slower.

**3. On the Unverified Claims of "Modify" and "Delete" Operations**

The authors repeatedly claim that NeuralDB is "easy to manage for supporting operations such as appending, modifying, and deleting". While appending is well-demonstrated, the "modifying" and "deleting" operations are completely unverified by experiments.

This is not a trivial claim. For example:

**Delete:** What happens when a $(k_i, r_i)$ pair is deleted from the database? Does the model revert to its original, pre-trained knowledge for that query? Or does it become confused and generate an error?
**Modify:** How is "modification" defined? Is it simply updating an existing residual $r_i$ for a key $k_i$? Or is it adding a new, conflicting fact about the same subject (which raises the question of key collision, a point not fully addressed in the paper)?

To validate these practical claims, could the authors provide a concrete case study? For instance: (1) Edit a fact. (2) Test that the edit is successful. (3) Delete the fact's $(k_i, r_i)$ pair from the NeuralDB. (4) Re-test the *same* prompt and report the model's output. Does it successfully revert to the original, pre-trained knowledge?

**Questions:**

See weaknesses

---

> ### Author Response · Authors · 2025-11-18
> **The first response to Reviewer HArh**
>
> Dear Reviewer HArh:
>
> We would like to sincerely thank you for your valuable and insightful comments. We are encouraged that you find our method smart, solid, easily applicable, and effective. Following your constructive comments, we have added the suggested experiments, and we would like to invite you to check these new results. All the results and discussions have been included in the revised manuscript, which we believe has greatly improved the quality of the paper.
>
> >W1.  The Selection of the Gating Hyperparameter $\gamma$
>
> Thank you for sharing your concern. We have added the  $\gamma$ ablation on GPT2-XL and GPTJ-6B  (Table 1), in addition to Llama-3 (Appendix I.3). Although all the evaluation metrics are usually monotonically changed as $\gamma$ increases,  **we observe a consistent trend across three models.** $\gamma = 0.65$ provides a good balance across metrics for three models.  Importantly, the generalization metric is monotonic increasing, while the specificity metric is monotonic decreasing. Therefore,  extremely low or high $gamma$ substantially degrades some metrics. We recommend selecting $gamma$ in [0.55, 0.65], which empirically has a good performance for all the metrics and three models.
>
> Table 1:  the $\gamma$ ablation on GPT2-XL and GPTJ-6B
>
> | Gamma | Efficacy ↑ | Efficacy ↑ | Generalization ↑ | Generalization ↑ | Specificity ↑ | Specificity ↑ | Fluency ↑ | Fluency ↑ |
> |:-----:|-----------:|-----------:|-----------------:|-----------------:|--------------:|--------------:|----------:|----------:|
> | Model |  GPTJ-6B   |  GPT2-XL   |     GPTJ-6B      |     GPT2-XL      |    GPTJ-6B    |    GPT2-XL    |  GPTJ-6B  |  GPT2-XL  |
> | 0.15  |    84.10   |    94.85   |       78.95      |       90.30      |     56.20     |     58.58     |  611.00   |  587.88   |
> | 0.40  |    97.55   |    99.70   |       97.02      |       98.38      |     63.92     |     65.66     |  600.47   |  611.20   |
> | 0.55  |    99.75   |    99.65   |       98.22      |       98.10      |     72.55     |     76.62     |  620.01   |  619.14   |
> | 0.65  |    99.80   |    99.80   |       97.20      |       94.60      |     74.10     |     80.00     |  621.50   |  619.80   |
> | 0.75  |    99.75   |    99.45   |       93.35      |       81.42      |     75.00     |     81.34     |  621.75   |  620.38   |
> | 0.90  |    98.55   |    96.70   |       57.38      |       30.00      |     76.68     |     82.57     |  621.35   |  620.30   |
>
>
> >W2. The True Cost of "Scalability" (Memory and Compute)
>
> Thank you for raising the question. We agree with your memory calculation. Our claim of “controllable overhead” refers to the 10k-fact setting reported in the main text, where the additional memory and latency are modest in practice.
>
> To our knowledge, we are the first to scale editing to 100k facts, and we now make this trade-off explicit. Our KV module functions as a vector database inside the FFN layer, so standard compression/indexing techniques (e.g., low-rank/SVD, quantization, and ANN retrieval) can reduce both memory and latency.
>
>  We have included the detailed results of running time in Table 2 for the ratio of extended latency across different numbers.  This has been added to the main paper (page 6) in the revised manuscript.
>
> Table 2: The ratio of additional latency compared with AlphaEdit
>
> | Total number of Edited facts              | 2k       | 4k       | 6k       | 10k        |  12k        |  16k       | 20k        |
> |----------------------------|:-----:|:-----:|:-----:|:-----:|:-----:|:-----:|:-----:|
> | The ratio of additional time | 0.65% | 1.65% | 1.67% | 1.7% | 2.29% | 3.69% | 5.55% |

---

> > ### Author Response · Authors · 2025-11-18
> > **The second response to Reviewer HArh**
> >
> > >W3. The Unverified Claims of "Modify" and "Delete" Operations
> >
> > Thank you for raising this valuable question.  Our knowledge editing is implemented using the KV database. Thus, we can modify and delete knowledge by modifying or removing entries therein. Following your suggestion, we now report both quantitative and qualitative evidence of the "Delete" Operation.
> >
> > **Setting.** We first edit 100 facts on GPT2-XL, then delete the keys and values for the even-indexed facts. This yields three models: (i) pre-edited, (ii) post-edited, and (iii) deleted-even (after removing edits for even facts only).
> >
> > **Quantitative.** **As shown in Table 3, performance on even facts for the deleted-even model matches the pre-edited model, while performance on odd facts is similar to the post-edited model.** This demonstrates selective reversibility (deletions roll back only the targeted facts) and retention (non-deleted facts remain effective).
> >
> > **Qualitative.** As suggested by the reviewer, we have added a concrete edit–delete-even case study. For the 4th edited fact:
> > >Question: “Autonomous University of Madrid, which is located in”, Target answer: “Sweden”.
> >
> > We report the answers of three models :
> >
> > - pre-edited model: “the city of Madrid, Spain”
> > - Post-edited model: “**Sweden**, has been working on a project called "The Future of the Human Body" for the past two years.”
> > - deleted-even model: “the city of Madrid, Spain”
> >
> > The answers to the deleted-even model successfully revert to the original answer to the pre-edited model.
> >
> > Table 3: The quantitative results of before editing, after editing, and after deleting odd edits
> >
> > | Group | Metric         | Pre-edited model | Post-edited model | Deleted-even model |
> > |:-----:|----------------|-----------:|------------:|-----------------:|
> > | Odd-indexed facts   | Efficacy       | 0.0        | 100.0       | 100.0            |
> > | Odd-indexed facts  | Generalization | 22.0       | 98.0        | 98.0             |
> > | Even-indexed facts   | Efficacy       | 0.0        | 100.0       | 0.0              |
> > | Even-indexed facts   | Generalization | 24.0       | 99.0        | 22.0             |

---

### Meta-Review · Area_Chair_kXYe · 2026-01-07

**Summary:**

This work proposes a knowledge editing approach that adds knowledge via KV database. The method is motivated by showing that existing methods can also be viewed in this formalism. Empirical evaluation shows that the method can edit up to 100K facts, which is more than previous work.
The reviewers appreciated the contribution and its ability to edit a large number of facts. The authors have addressed most of the concerns raised. by the two lower-scoring reviewers.
One important thing that the authors should be more transparent about is that their edits increase the size of the model, and thus it isn't a fair comparison to compare it to editing methods that do not. In that sense "Improving Sequential Model Editing with Fact Retrieval" which one reviewer mentioned is the most relevant baselines. The authors mention that this paper does "external" retrieval, but this distinction should be clarified further.  In that respect the authors should also definitely cite "Adaptable and Interpretable Neural Memory
Over Symbolic Knowledge" and other similar work which also proposes Key-Value type edits.
Another concern is that because the Key-Value edit is stored externally, it may not fit in the way the model stores facts generally, and thus it's possible that more complex queries may not generalize well. As one reviewer mentioned, the paper would benefit from evaluating on MQUAKE or RippleEdits.

**Reviewer Concerns:**

Reviewers had questions about parameter tuning, scalability, and related work and these have been addressed. Outstanding issues are generalization to more complex benchmarks such as MQuake and RippleEdits.

**Reviewer Scores:**

Two reviewers updated their scores and acknowledge this in the discussion.

---

### Decision · Program_Chairs · 2026-01-26

Accept (Poster)